# Controllable Data Generation with Hierarchical Neural Representations

## Abstract

Implicit Neural Representations (INRs) represent data as continuous functions using the parameters of a neural network, where data information is encoded in the parameter space. Therefore, modeling the distribution of such parameters is crucial for building generalizable INRs. Existing approaches learn a joint distribution of these parameters via a latent vector to generate new data, but such a flat latent often fails to capture the inherent hierarchical structure of the parameter space, leading to entangled data semantics and limited control over the generation process. Here, we propose a **C**ontrollable **H**ierarchical **I**mplicit **N**eural **R**epresentation (**CHINR**) framework, which explicitly models conditional dependencies across layers in the parameter space. Our method consists of two stages: In Stage-1, we construct a Layers-of-Experts (LoE) network, where each layer modulates distinct semantics through a unique latent vector, enabling disentangled and expressive representations. In Stage-2, we introduce a Hierarchical Controllable Diffusion Model (HCDM) to capture conditional dependencies across layers, allowing for controllable and hierarchical data generation at various semantic granularities. Extensive experiments on CelebA-HQ, ShapeNet, SRN-Cars, and AMASS datasets demonstrate that CHINR improves generalizability and offers flexible hierarchical control over the generated content.

## 1 Introduction

Implicit Neural Representations (INRs) are powerful tools to represent complex data as continuous functions with neural network parameters (Tancik et al., 2020; Mildenhall et al., 2021; Dupont et al., 2022a; You et al., 2024). This functional representation is agnostic to the underlying data modality, making INRs a universal way to represent diverse data types, including audio, images, and 3D volumes. Unlike conventional discrete data structures with explicit formats, INRs implicitly encode data in the parameter space. This allows for more detailed, compact, and flexible representations. Recent advancements in the quality and efficiency of INRs greatly enhanced their performance across various applications, such as image fitting (Sitzmann et al., 2020; Dupont et al., 2021a), video compression (Chen et al., 2021a; 2023; Li et al., 2022), shape modeling (Zhao et al., 2022; Michalkiewicz et al., 2019), novel view synthesis (Mildenhall et al., 2021), and beyond.

The generalization ability of INRs, however, remains limited, since INRs are typically trained to overfit individual data instances. Keeping in mind that all data information is embedded in INR's parameters, the parameter space acts as a hidden manifold that captures the structures and variations of data. Building on this concept, a recent line of work (Perez et al., 2018; Chan et al., 2021; Dupont et al., 2022a; You et al., 2024) has focused on modeling the parameter distribution to generalize INRs. A common approach is to adopt a two-stage framework. First, a collection of INRs is fitted to each data instance. Given a collection of data $\{(\mathbf{x}, \mathbf{f})\}$, each INR $f_{\boldsymbol{\theta}} : \mathcal{X} \rightarrow \mathcal{F}$ is learned by finding weight $\boldsymbol{\theta}$ to minimize the distance between $f_{\boldsymbol{\theta}}(\mathbf{x})$ and $\mathbf{f} \in \mathcal{F}$. $\mathbf{x} \in \mathcal{X}$ denotes coordinates (*e.g.* pixel locations) and $\mathbf{f} \in \mathcal{F}$ denotes signals (*e.g.* RGB values). Second, the weight distribution $p(\boldsymbol{\theta})$ is learned through generative models. Since the raw INR parameters are intractably high-dimensional, a condensed latent vector $\mathbf{h}$ is often used to represent each INR via modulation (Dupont et al., 2022a). This simplifies the task of learning $p(\boldsymbol{\theta})$ by transforming it into the surrogate task of learning $p(\mathbf{h})$, which exists in a more manageable lower-dimensional space.

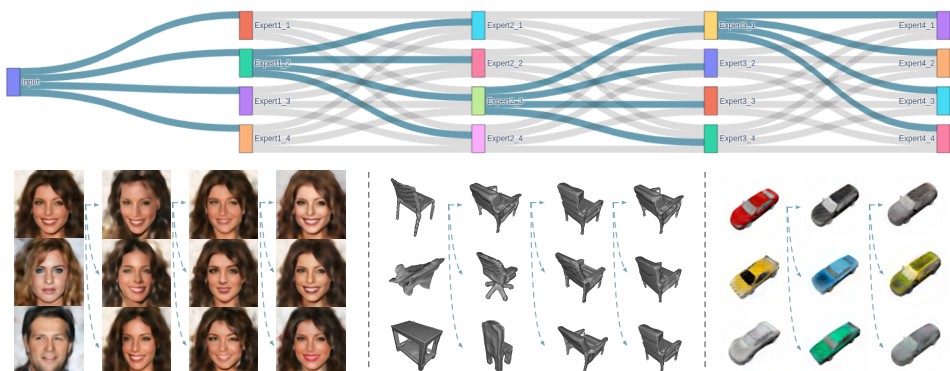

Figure 1: Hierarchical generation of universal data modality. *Top*: expanded representational capabilities of a Layers-of-Experts INR model. *Bottom*: aligning these capabilities with hierarchical data patterns enables precise control over the generation process. Each column presents generated samples resulting from a divergence in routing at a specific layer, with arrows indicating the shared routing in the preceding layers.

Despite the various modeling approaches, these methods learn the joint distribution of all parameters of an INR, treating each INR's parameters indiscriminately via a flattened latent. This approach overlooks the inherent hierarchical structure of INRs, particularly in multilayer perceptron (MLP)-based architectures such as SIREN (Sitzmann et al., 2020), where each layer's representation capacity depends conditionally on the previous layers. As a result, these methods fail to capture the hierarchical patterns in the data. Taking the facial image as an example, hierarchical semantics refer to progressively detailed facial characteristics, such as overall facial shape, expression, and shape of eyes. Meanwhile, the parameters of INRs exhibit layer-wise expanded representation capacity (Yüce et al., 2022), determined by its hierarchical structure. However, learning the joint distribution with flattened parameters ignores the correspondences between the INRs' expanded representation capacity across layers and the hierarchical patterns in the data. This misalignment brings two challenges: (1) The generalizability of INRs to unseen data is impaired. The joint distribution learning encodes the entangled semantics together into one flat latent, where the co-occurrence of certain semantics is inevitable. Therefore, the diversity of the generated data is limited. (2) Control over the generated content is limited. While generative models, such as latent diffusion, are employed with INRs to generate new data, they cannot link the sampled noise to the expected semantics in the output.

To address these challenges, we propose a novel approach that models the hierarchical structure of INRs, promoting layer-wise control in the generation process. Our method starts by training a collection of INRs on a dataset. Each layer of an INR is parameterized as a Mixture-of-Experts (MoE) layer to increase expressivity, where a set of expert weights and a latent vector are learned. The experts at each layer are shared across the dataset, while the latents are data-specific, routing the data flow and modulating the contribution of experts. Layers of MoE are cascaded to form an INR, which we call a Layers-of-Experts (LoE) network. Consequently, a LoE with $L$ layers will have $L$ latents adapted to the fitted data, effectively capturing and relating its complex patterns with layers of latents. By modeling the conditional dependencies of these layer-wise latents with our proposed hierarchical controllable diffusion model (HCDM), we maintain the hierarchical structure of INRs. This unlocks a controllable generation process, aligning the layers in INRs with the hierarchies of data semantics for the first time. As illustrated in Fig. 1, the data flow for a generation process resembles a tree-like structure: the routing in the next layer is constrained by the paths in previous layers, allowing full control of where a different routing strategy should be explored. Early deviations in routing lead to significant semantic differences in the generated content, while a later deviation results in minor differences in details.

The contributions of our paper are summarized as follows:

- We model the INR as a LoE framework, where each layer includes multiple experts to improve expressivity. Additionally, the layer-wise latent structure aligns the inherent hierarchy of data semantics with INR's expanded representation capacity.

- We are the first to model the hierarchical and conditional dependencies within INR parameters with the proposed HCDM, which enables hierarchical control over the generated data semantics at different levels of granularity.

- We offer theoretical analysis and empirical evidence to show the inherent connection between the hierarchy of INR parameters and data semantics, validated through extensive experiments across various modalities.

## 2 BACKGROUND

In this section, we introduce INRs and their generalization ability, while highlighting their connections to our proposed approach. We also analyze the inherent hierarchy in INR's parameters.

### 2.1 IMPLICIT NEURAL REPRESENTATION AND GENERATIVE INRS

Implicit Neural Representations (INRs) parameterize data such as audio, images, video, and 3D voxels as mappings from coordinates to signals, enabling a unified framework for various data modalities (Genova et al., 2019a;b; Xie et al., 2022). Remarkable progress has been made to enhance the representation quality, efficiency and compactness for audio (Zuiderveld et al., 2021; Luo et al., 2022; Su et al., 2022; Lanzendörfer & Wattenhofer, 2023), images (Sitzmann et al., 2020; Fathony et al., 2020; Chen et al., 2021b; Xu et al., 2022; Saragadam et al., 2023), 3D contents (Mildenhall et al., 2021; Barron et al., 2021; Tiwari et al., 2022; Ortiz et al., 2022; Zhao et al., 2022; Ruan et al., 2024), and videos (Chen et al., 2021a; Li et al., 2022; Yan et al., 2024). Despite performing well on different modalities, INRs struggle to generalize to multiple and unseen data, as each instance is typically overfitted with a separate MLP. To address this, two key strategies have emerged: (1) learning content-specific input features (Yu et al., 2021; Hu et al., 2023; Lazova et al., 2023)(Xu et al., 2024) and (2) modulating or customizing network parameters with latents or hypernetworks (Mehta et al., 2021; Wang et al., 2022; Dupont et al., 2022b; Kim et al., 2023)(Xu et al., 2024). Generative models (Goodfellow et al., 2014; Ho et al., 2020) further extend INR's capability to generate new data. GRAF (Schwarz et al., 2020) and GIRAFFE (Niemeyer & Geiger, 2021) generate shape and appearance codes from noise, which are combined with coordinates to construct scenes. Erkoç et al. (2023) use a diffusion model to generate INR weights. Dupont et al. (2021b); Du et al. (2021); Koyuncu et al. (2023) train hyper-networks to generate INR parameters. Dupont et al. (2022a); Bauer et al. (2023); Park et al. (2024) employ a two-stage framework to learn the distribution of latents that map to or modulate INRs, and generate new content by sampling in the latent space. mNIF (You et al., 2024) further enhances the expressivity of INR via model averaging. These methods essentially model the distribution of INR parameters $p(\boldsymbol{\theta})$ by learning latent distributions $p(\boldsymbol{h})$, but fail to capture the layer-wise hierarchical structure of INR parameters, limiting their ability to accurately model distributions and control generation, which will be discussed in Sec. 2.2. Building on the latent modulation approach, we introduce a hierarchical controllable diffusion model, capturing dependencies between layer-wise latents for improved generalization and control.

### 2.2 HIERARCHY ANALYSIS OF INR

In this section, we review the INR architecture and analyze its inherent hierarchical representation ability. Using the SIREN (Sitzmann et al., 2020) as an example, a two-layer SIREN is generally formulated as:

$$f_{\boldsymbol{\theta}}(\mathbf{x}) = \mathbf{W}_2 \sin(\mathbf{W}_1 \cdot \gamma(\mathbf{x})), \boldsymbol{\theta} = [\mathbf{W}_1, \mathbf{W}_2], \quad (1)$$

where $\gamma(\mathbf{x}) = \sin(\Omega \cdot \mathbf{x}), \Omega \in \mathbb{R}^{c_1 \times c_{in}}$ denotes positional embedding of coordinates $\mathbf{x}$, $\mathbf{W}_2 \in \mathbb{R}^{c_{out} \times c_2}$, $\mathbf{W}_1 \in \mathbb{R}^{c_2 \times c_1}$ denote the parameters of each layer. The bias is omitted for simplification. From the perspective of Fourier-Transformation, the input frequency domain $\Omega$ is composed of $c_1$ frequency basis, $\Omega = [\Omega_0, \Omega_2, \cdots, \Omega_{c_1-1}]$. According to the Tancik et al. (2020) and Yüce et al. (2022), an MLP layer with periodic activation $\sin(\cdot)$ only expands the input frequency basis in a sparse and limited bandwidth. The equation 1 can be reformulated as:

$$f_{\boldsymbol{\theta}}(\mathbf{x}) = \sum_{w' \in \mathcal{H}(\Omega)} \alpha_{w'} \sin(w' \cdot \mathbf{x}), \quad \alpha_{w'} \propto \mathbf{W}_2 \cdot \prod_{c=0}^{c_1-1} \mathcal{J}_{s_c}(\mathbf{W}_{1[\cdot,c]})$$

$$\mathcal{H}(\Omega) \subseteq \{\sum_{c=0}^{c_1-1} \beta_c \Omega_c | \beta_c \in \mathbb{Z} \ \& \ \sum_{c=0}^{c_1-1} \beta_c \leq K\}, \quad (2)$$

Figure 2: The proposed CHINR consists of two stages. In Stage-1, a Layer-of-Experts (LoE) model is used to represent large-scale data, utilizing instance-specific latents and shared experts. Stage-2 introduces a Hierarchical Controllable Diffusion Model (HCDM) to learn the layer-wise conditional distributions of the latents.

where $\mathcal{J}_{s_c}$ denotes a Bessel function, $\mathbf{W}_{1[\cdot,c]}$ denotes the column $c$ of $\mathbf{W}_1$. The equation 2 reveals the properties of each $\sin(\cdot)$ activated INR layer in two aspects. First, the output spectrum of layer 2, i.e. $\alpha_{w'}$, is dependent on the spectrum of layer 1, determined by $\mathbf{W}_1$; Second, the output frequency domain $\mathcal{H}(\Omega)$ is sparse since $\beta_c$ is an integer, so $\mathcal{H}(\Omega)$ only covers sparse frequency space spanned by the basis $\{\beta_c \Omega_c\}$. These suggest that INR layers' spectrum and frequency basis inherently exhibit a sparse and hierarchical structure, encoded by $\boldsymbol{\theta}$, which extends to their representation ability. Latent-modulation approaches like mNIF (You et al., 2024), which model the parameter distribution $p(\boldsymbol{\theta})$ with the surrogate task of modeling the latent distribution $p(\boldsymbol{h})$, overlook the hierarchy in $\boldsymbol{h}$ transferred from $\boldsymbol{\theta}$. Ignoring this hierarchy leads to reduced expressivity and generalizability, and limited control over the generation process.

## 3 PROPOSED METHOD

Our method uses a two-stage framework to align the hierarchy of data semantic and INR's expanded representation ability, as shown in Fig. 2. In the first stage, we train individual INRs to fit a target dataset. In the second stage, we use generative models to learn the weight distribution for generating new data. Directly modeling the INR weight distribution brings three challenges: (1) independently trained INRs make it hard to extract shared information for distribution learning, (2) the high dimensionality of raw weights makes distribution modeling highly challenging, and (3) it ignores the hierarchical structure of INRs. To address these issues, we configure the INR as a Layers-of-Experts (LoE) network, where each layer contains a set of shared expert weights and an instance-specific latent vector. As shown in the left of Fig. 2, the inference process builds each INR layer by layer, combining the experts with the corresponding latent vector. This structure captures shared information through the experts, simplifies distribution learning by focusing on the latents, and explicitly models the conditional dependencies within the hierarchical structure of INRs. In the following sections, we first define the LoE structure and the learning task, followed by detailed explanations of the two stages.

### 3.1 PROBLEM STATEMENT

Suppose an INR $f_{\boldsymbol{\theta}}$ has $L$ layers. For layer $l$, we learn a collection of $K$ cross-data shared expert weights $\boldsymbol{\theta}^l = \{\boldsymbol{\theta}_1^l, \boldsymbol{\theta}_2^l, \cdots, \boldsymbol{\theta}_K^l\}$ (each being a fully connected layer) and a unique latent $\mathbf{h}^l \in \mathbb{R}^H$ for each data instance. At inference, the operation for the LoE INR at layer $l$ is:

$$\boldsymbol{y}^{l+1} = \sin(\omega_0 \cdot (\bar{\boldsymbol{\theta}}^l \cdot \boldsymbol{y}^l)), \quad \bar{\boldsymbol{\theta}}^l = \sum_{n=1}^{K} \boldsymbol{\theta}_k^l \cdot \alpha_k^l,$$

$$[\alpha_1^l, \alpha_2^l, \cdots, \alpha_K^l]^\top = \boldsymbol{\alpha}^l = g_{\boldsymbol{\phi}}(\mathbf{h}^l), \tag{3}$$

where $\boldsymbol{y}$ represents each layer's output, $\omega_0$ is a constant factor, and $\bar{\boldsymbol{\theta}}^l$ denotes instance-specific parameters at layer $l$, modulated by a *gating vector* $\boldsymbol{\alpha}^l$, which is computed by the gating module $g_{\boldsymbol{\phi}}(\cdot)$. $\mathbf{h}^l$ denotes the $l_{th}$ layer of latent $\mathbf{h}$. Compared with directly learning the gating vectors, the function $g_{\boldsymbol{\phi}}(\cdot)$ allows for a more compact latent that benefits distribution learning. By modulating

the contribution of experts via latents, each layer gains the flexibility to adapt to individual data samples using a shared basis. Since $L$ layers are cascaded to form the final INR, its expressive capacity is significantly enhanced through the integrated contributions across layers.

In Stage-1, we optimize the shared network parameters denoted as $\boldsymbol{\theta} = \{\boldsymbol{\theta}^1, \boldsymbol{\theta}^2, \cdots, \boldsymbol{\theta}^L, \phi\}$, and fully characterize each instance-specific INR by layer-wise stacked latents $\mathbf{h} = [\mathbf{h}^1, \mathbf{h}^2, \cdots, \mathbf{h}^L] \in \mathbb{R}^{H \times L}$. The layer-wise latent structure enables hierarchical modeling of INR parameters, which aligns with the inherent hierarchy of data semantics, allowing for layer-wise dependency modeling in Stage-2, and controllable data generation.

## 3.2 STAGE-1: LEARNING A DATASET OF LoE INRS

Similar to Functa (Dupont et al., 2022a) and mNIF (You et al., 2024), we use meta-learning and auto-decoding to train both the data-specific latents $\mathbf{h}$ and the shared parameters $\boldsymbol{\theta}$ for the LoE INR during Stage-1. For meta-learning, we adopt an interleaved training procedure inspired by CAVIA (Zintgraf et al., 2019), where the experts and latents are updated alternately in separate training loops. In the inner loop, we fix $\boldsymbol{\theta}$ and adapt the latents $\mathbf{h}$ to data samples. Within each inner loop, $\mathbf{h}$ is first randomly initialized around zero and then updated for a few steps. In the outer loop, $\boldsymbol{\theta}$ is optimized based on the updated $\mathbf{h}$. This ensures each data-specific latent can be effectively learned within a few iterations, encouraging faster convergence and adaptation to new data, which is essential for distribution modeling and generalization in Stage-2. In the case of auto-decoding, we jointly optimize all parameters, maintaining a latent bank for the dataset and updating the sampled batch of latents in each iteration. Unlike meta-learning, auto-decoding does not require second-order derivatives, making it more computationally efficient. Due to this efficiency, we apply auto-decoding specifically for NeRF training.

In both approaches, each data-specific latent $\mathbf{h}$ consists of $L$ components separately modulating the $L$ layers in the LoE INR. This setup facilitates conditional distribution modeling for Stage-2, as opposed to learning their joint distribution like mNIF and Functa, enabling hierarchical and controllable generation.

## 3.3 STAGE-2: CONDITIONAL DISTRIBUTION LEARNING FOR GENERALIZABLE INRS

Given a collection of learned latents $\mathcal{H} = \{\mathbf{h}_1, \mathbf{h}_2, \cdots \mathbf{h}_N | \mathbf{h}_n \in \mathbb{R}^{H \times L}\}$, where $N$ denotes the number of data instances, $L$ the number of layers, and $H$ the dimension of each layer of the latent, the Stage-2 aims at learning the distribution of latent $p(\mathbf{h})$. Instead of blindly modeling the joint distribution, we reformulate $p(\mathbf{h})$ as follows:

$$p(\mathbf{h}) = p(\mathbf{h}^1, \mathbf{h}^2, \cdots, \mathbf{h}^L) = p(\mathbf{h}^1) \prod_{l=2}^{L} p(\mathbf{h}^l | \mathbf{h}^{<l}), \qquad (4)$$

where $p(\mathbf{h}^{<l}) = p(\mathbf{h}^1, \cdots, \mathbf{h}^{l-1})$ denotes the joint probability of first $l-1$ layers. We design a hierarchical conditional diffusion model (HCDM) to learn the conditional dependency $p(\mathbf{h}^l | \mathbf{h}^{<l})$ in equation 4. As shown in Fig. 2, Stage-2 illustrates the architecture of the HCDM, which includes a forward process and a backward process.

### 3.3.1 FORWARD PROCESS

We initialize the $\mathbf{h}$ at step 0 as $\mathbf{h}_0 = [\mathbf{h}_0^1, \cdots, \mathbf{h}_0^L]$ with a conditional chain of length $L$, the forward process for each layer $\mathbf{h}^l$ is formulated as:

$$q(\mathbf{h}_{1:T}^l | \mathbf{h}_0^l) := \prod_{t=1}^{T} q(\mathbf{h}_t^l | \mathbf{h}_{t-1}^l), \quad q(\mathbf{h}_t^l | \mathbf{h}_{t-1}^l) := \mathcal{N}(\mathbf{h}_t^l; \sqrt{1-\beta_t} \mathbf{h}_{t-1}^l, \beta_t \mathbf{I}), \qquad (5)$$

where $q(\mathbf{h}_t^l | \mathbf{h}_{t-1}^l)$ denotes the posterior distribution of the $\mathbf{h}_t^l$ given $\mathbf{h}_{t-1}^l$, $T$ denotes the number of diffusion steps. $\beta_1, \cdots, \beta_T$ denotes the variance schedule of the added Gaussian noise $\mathcal{N}(\cdot)$. By the forward process, the noise sample $\mathbf{h}_T = [\mathbf{h}_T^1, \cdots, \mathbf{h}_T^L]$ is generated from $\mathbf{h}_0 = [\mathbf{h}_0^1, \cdots, \mathbf{h}_0^L]$.

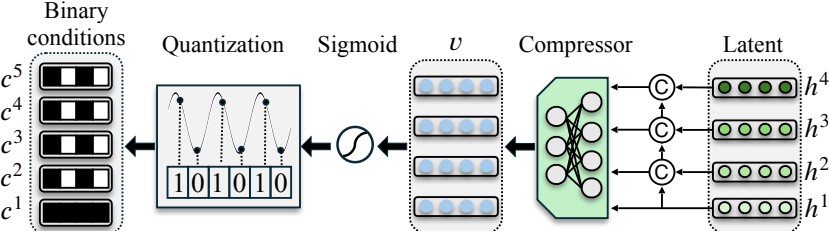

Figure 3: Condition formation. Layers of latent $\mathbf{h}$ are first concatenated and then compressed. The compressed tensors are binarized to generate low-dimensional binary conditions.

### 3.3.2 BACKWARD PROCESS

The backward process models the prior distribution defined by equation 4. To model the hierarchical structure, the backward process should express the conditional dependency $p(\mathbf{h}^l|\mathbf{h}^{<l})$ for all $L$ components. Therefore, we take the $\mathbf{h}^{<l}$ as condition, and generate $\mathbf{h}^l$ for $l = 1, \cdots L$ iteratively. To understand how to process $\mathbf{h}^{<l}$ as a condition, we first explain the details of condition formation.

**Condition Formation.** To generate a latent $\mathbf{h} = [\mathbf{h}^1, \cdots, \mathbf{h}^L]$, the condition formation prepares for each $\mathbf{h}^l$ a condition vector $\mathbf{c}^l$ encapsulating $\mathbf{h}^{<l}$. Since $\mathbf{h}$ resides on a low-dimensional manifold (e.g., 64), the condition $\mathbf{c}^l$ should contain less information to prevent the HCDM from memorizing all one-to-one mappings from $\mathbf{h}^{<l}$ to $\mathbf{h}^l$, where $p(\mathbf{h})$ inevitably degenerates into $p(\mathbf{h}^1)$. Therefore, we embed the conditions $\mathbf{h}^{<l}$ into a lower-dimensional binary vector. Fig. 3 shows the framework of condition formation, which includes two steps. (1) All $\mathbf{h}^{<l}$ are concatenated and compressed with a compressor $\mathbf{W}$ to get a compressed tensor $\mathbf{v}^l \in \mathbb{R}^C$: $\mathbf{v}^l = \mathbf{W} \cdot \text{concat}(\mathbf{h}^1, \cdots, \mathbf{h}^L), \mathbf{W} \in \mathbb{R}^{C \times HL}, \mathbf{h}^j \in \mathbb{R}^H$, where $C$ denotes the compressed dimension. To match the dimensions of $\mathbf{W}$ with the concatenated tensors, we set the values of the unused portion to zero. (2) Given the compressed tensor $\mathbf{v}^l$, a binarized condition tensor $\mathbf{c}^l$ is obtained by: $\mathbf{c}^l = \mathcal{Q}(\sigma(\mathbf{v}^l))$, where $\mathcal{Q}(\cdot)$ denotes the binarization operation, $\sigma$ denotes the "Sigmoid" function. For $\mathbf{c}^1$, we set it as a zero tensor since $\mathbf{h}^1$ has no condition. Now we get the condition $\mathbf{c}^1, \cdots, \mathbf{c}^L$.

**Hierarchical generation.** With the condition $\mathbf{c} = [\mathbf{c}^1, \cdots, \mathbf{c}^L]$, noise sample $\mathbf{h}_T = [\mathbf{h}_T^1, \cdots, \mathbf{h}_T^L]$, time step $t$, we are all prepared to generate a complete latent $\mathbf{h}_0$. To generate a component $\mathbf{h}_0^l$, the backward process is formulated as: $p(\mathbf{h}_{0:T}^l|\mathbf{c}^l) = p(\mathbf{h}_T^l)\prod_{t=1}^T p(\mathbf{h}_{t-1}^l|\mathbf{h}_t^l, \mathbf{c}^l)$. It is noted that a complete backward process, generating a sample from $p(\mathbf{h}_0^l|\mathbf{c}^l)$, is exactly the implementation of $p(\mathbf{h}^l|\mathbf{h}^{<l})$ in equation 4, where $\mathbf{c}^l$ corresponds to $\mathbf{h}^{<l}$. We adopt a UNet (Ronneberger et al., 2015) $\mu_{\boldsymbol{\theta}}$ as in Song et al. (2020); Ho et al. (2020) to implement $p(\mathbf{h}_{t-1}^l|\mathbf{h}_t^l, \mathbf{c}^l)$:

$$p(\mathbf{h}_{t-1}^l|\mathbf{h}_t^l, \mathbf{c}^l) = \mathcal{N}(\mathbf{h}_{t-1}^l : \boldsymbol{\epsilon}_{\boldsymbol{\theta}}(\mathbf{h}_t^l, t, \mathbf{c}^l), \boldsymbol{\Sigma}(t))$$

$$\boldsymbol{\epsilon}_{\boldsymbol{\theta}}(\mathbf{h}_t^l, t, \mathbf{c}^l) = \frac{\sqrt{a_t}(1 - \bar{a}_{t-1})\mathbf{h}_t^l + \sqrt{\bar{a}_{t-1}(1 - a_t)}\mu_{\boldsymbol{\theta}}(\mathbf{h}_t^l, t, \mathbf{c}^l)}{1 - \bar{a}_t} \tag{6}$$

$$a_t = 1 - \beta_t, \quad \bar{a}_t = \prod_{i=1}^t a_i, \quad \boldsymbol{\Sigma}(t) = \frac{(1 - a_t)(1 - \bar{a}_{t-1})}{1 - \bar{a}_t}\mathbf{I}.$$

Each backward process starts from a Gaussian noise sample $\mathbf{h}_T^l$ and generates a latent component $\mathbf{h}_0^l$ with condition $\mathbf{c}^l$. By iteratively sampling from $p(\mathbf{h}^1)$ and $p(\mathbf{h}^l|\mathbf{h}^{<l})$ with $l = 2, \cdots L$, a complete latent $\mathbf{h}_0 = [\mathbf{h}_0^1, \cdots, \mathbf{h}_0^L]$ is generated. The final objective is: $\mathcal{L}_{hcdm} = \mathbb{E}_{\mathcal{H},t,l}[\lambda||\epsilon, \boldsymbol{\epsilon}_{\boldsymbol{\theta}}(\mathbf{h}_t^l, t, \mathbf{c}^l)||^2]$, where $\mathcal{H}$ denotes all latents, $\epsilon$ denotes Gaussian sample from $\mathcal{N}(\mathbf{0}, \mathbf{I})$, and $\lambda$ is a constant coefficient. For inference, we first generate $\mathbf{h}^1$ from Gaussian noise, then perform a chain of conditional sampling from $p(\mathbf{h}^l|\mathbf{h}^{<l})$ until we get the complete latent $\mathbf{h}$ used by the LoE to generate new data.

## 4 EXPERIMENTS

In this section, we begin by introducing the datasets and evaluation criteria. Then, we provide quantitative and qualitative results to demonstrate the efficacy of our proposed framework. Next, we highlight the controllability of the INR generation process, enabled by conditional dependency modeling and hierarchical sampling. Furthermore, we analyze the latent space and illustrate how

data semantics are embedded within INR's weight space hierarchically. We show that this structure grants the INR a *compositional* property across layers. Finally, we conduct ablation studies to show the importance of conditional dependency modeling and the functionality of binary conditions.

In our experiments, both Stage-1 and Stage-2 are trained and evaluated on the CelebA-HQ $64^2$ (Karras, 2017), ShapeNet $64^3$ (Chang et al., 2015), SRN-Cars (Sitzmann et al., 2019), and AMASS (Mahmood et al., 2019) datasets. All experiments are implemented in Pytorch and run on a single Nvidia RTX3090 GPU. For evaluation metrics, we use peak signal-to-noise ratio (PSNR), structural similarity (SSIM) (Wang et al., 2004), and accuracy to assess Stage-1 reconstruction, and Fréchet inception distance (FID) (Heusel et al., 2017), coverage, and maximum mean discrepancy (MMD) (Achlioptas et al., 2018) to evaluate Stage-2 generation performance. Further details about implementation can be found in the Appendix.

Table 1: Quantitative results on CelebA-HQ, ShapeNet, and SRN-Cars.

| Model | CelebA-HQ | | | ShapeNet | | | | SRN-Cars | | |
| | Reconstruction | | Generation | Reconstruction | | Generation | | Reconstruction | | Generation |
| | PSNR↑ | SSIM↑ | FID↓ | PSNR↑ | Accuracy↑ | Coverage↑ | MMD↓ | PSNR↑ | SSIM↑ | FID↓ |
|---|---|---|---|---|---|---|---|---|---|---|
| Functa (Dupont et al., 2022a) | 26.6 | 0.801 | 40.4 | - | - | - | - | 24.2 | 0.739 | 80.3 |
| GEM (Du et al., 2021) | - | - | 30.4 | 21.3 | - | 0.409 | 0.0014 | - | - | - |
| GASP (Dupont et al., 2021b) | - | - | 13.5 | 16.5 | - | 0.341 | 0.0021 | - | - | - |
| mNIF (You et al., 2024) | 34.5 | 0.958 | 13.2 | 21.4 | 0.972 | 0.437 | 0.0013 | 25.9 | 0.758 | 79.5 |
| CHINR | 34.9 | 0.963 | 13.4 | 22.3 | 0.988 | 0.441 | 0.0011 | 26.2 | 0.772 | 77.9 |

## 4.1 QUANTITATIVE AND QUALITATIVE RESULTS

Table 1 presents the reconstruction and generation performance of the proposed CHINR compared with baseline methods on different modalities. Our model outperforms existing methods on most datasets. In Stage-1, it achieves the highest reconstruction PSNR, thanks to the expanded representation capacity of LoE and layer-wise latent learning. In Stage-2, it demonstrates superior generalization, highlighting the effectiveness of hierarchical conditional modeling in capturing the diverse data semantics. On CelebA-HQ, our model achieves the second-lowest FID score, slightly behind mNIF. However, we observe that mNIF tends to generate images resembling those from the training set. To verify this, we perform retrieval for $1,000$ generated samples and compute the average of their minimum L2 distance to the training set images. A lower value indicates a higher chance of "memorizing" the training set. mNIF obtains a value of $6.243$ whereas CHINR obtains $15.971$, indicating that our model generalizes better by generating new images that differ more noticeably from the training data. Examples of retrieval results are provided in the Appendix.

Fig. 4 displays uncurated samples generated by our model compared to existing methods. With the proposed HCDM, our model generates high-quality samples with rich details. More results including the AMASS dataset are provided in the Appendix.

## 4.2 HIERARCHICAL CONTROLLABLE GENERATION

In this section, we show the hierarchical controllable generation with conditional sampling. Since the proposed HCDM successfully models the conditional dependencies of the modulation latents, we can conduct a chain of conditional samplings to achieve hierarchical control over semantics at different granularities in the generated contents. Specifically, to sample a latent $\mathbf{h} = [\mathbf{h}^1, \cdots, \mathbf{h}^5]$, we begin by sampling $\mathbf{h}^1$ with HCDM taking a Gaussian noise as input. Next, we sample $\mathbf{h}^2$, conditioned on the binary vector generated by $\mathbf{h}^1$. This process continues layer by layer until $\mathbf{h}^5$ is generated, forming a complete latent $\mathbf{h}$. This layer-wise latent then modulates the LoE to render output such as images, point clouds, or NeRF renderings. Fig. 5 illustrates this hierarchical control on two modalities. For each modality, we show two samples generated through the full chain of conditional sampling ($\mathbf{h}^1 \cdots \mathbf{h}^5$) in the first column. In the second column, we fix their $\mathbf{h}^1$ from the first column and sample the remaining parts ($\mathbf{h}^2 \cdots \mathbf{h}^5$). In the third column, we fix both $\mathbf{h}^1$ and $\mathbf{h}^2$ as in the second column and conditionally sample the rest. This progressive fixation allows us to control the finer details of the generated output.

As shown Fig. 5, two different chains of conditional sampling, each starting from a different initial $\mathbf{h}^1$, result in highly different data semantics (first column). For the facial images, when $\mathbf{h}^1$ is fixed, the generated samples exhibit similar overall contours but different contents, i.e., facial features, facial orientation, and hairstyles (second column). When $\mathbf{h}^1$ and $\mathbf{h}^2$ are fixed, variations occur

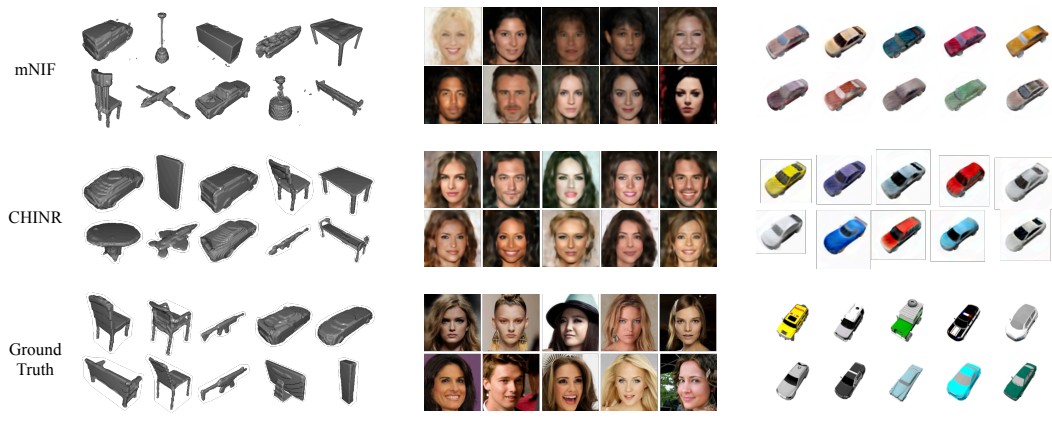

(a) ShapeNet        (b) CelebA-HQ        (c) SRN-Cars

Figure 4: Uncurated generations for ShapeNet, CelebA-HQ, and SRN-Cars datasets.

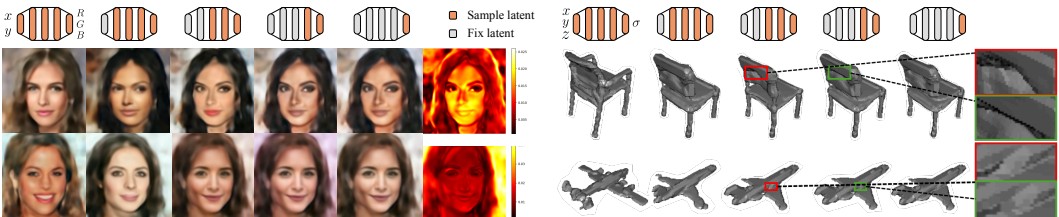

Figure 5: Controllable hierarchical generation by progressively fixing the layer-wise latent.

primarily in facial details, such as eye shape and hair color (third column). Fixing the first three parts (fourth column) results in changes limited to skin tone. Lastly, $\mathbf{h}^5$ influences global properties like foreground or background color, as highlighted by the heat maps in the last column. Similarly, for voxels, the hierarchy control is manifested in category (e.g., chair and plane), object parts (e.g., armrest and back of the chair), and finer details of specific parts (e.g., structure of the chair back).

This hierarchical control via conditional sampling demonstrates that HCDM effectively captures conditional dependencies of parameters, aligned with the hierarchy of data semantics. It also offers practical benefits in digital content creation, allowing creators to fine-tune specific details in their work. Additional examples of controllable generation are provided in the Appendix.

### 4.3 ANALYSIS

In this section, we unravel the reason behind the success of hierarchical controllable generation. We show that each layer's latent modulates disentangled semantics, giving our LoE model a compositional property. Moreover, the layer-wise conditional dependency modeling successfully captures the hierarchy of data semantics, ensuring the composed semantics are compatible across layers.

### 4.3.1 LATENT COMPOSITION

We explore the controllability of our method through latent composition and find that the learned latents are layer-wise compositional. Given two latents $\mathbf{h}_1 = [\mathbf{h}_1^1, \cdots, \mathbf{h}_1^5], \mathbf{h}_2 = [\mathbf{h}_2^1, \cdots, \mathbf{h}_2^5]$, exchanging a specific part, e.g. $\mathbf{h}_1^2$ and $\mathbf{h}_2^2$, results in the corresponding semantic changes in the generated content. As shown in Fig. 6, we randomly sample 9 latents from HCDM, trained on CelebA-HQ, and render the images in the first row. These faces exhibit diverse characteristics including expressions, hairstyles, facial orientations, skin tones, and background colors. In the second row, we replace the second layer's latent of the first sample with that of the other 8 samples, while keeping the rest latents unchanged. It's clear that the facial orientation, hairstyles, and background colors change accordingly, while the facial features remain the same. This indicates that the second part of the latent encodes these specific semantics disentangled from other semantics, laying the foundation for the controllable hierarchical generation. In the third row, when the third layer's latent is replaced, we observe that only the facial features are swapped, while other characteristics, such

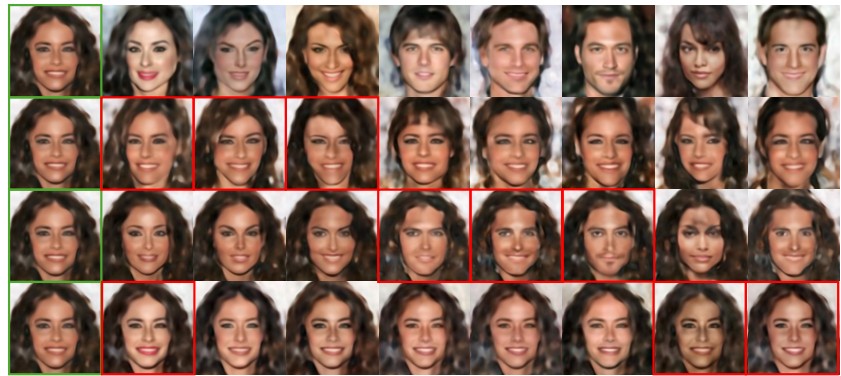

Figure 6: Latent composition. The first row presents 9 randomly sampled images from Stage-2. The second to the fourth rows present the images where the second to the fourth parts of the first sample's latent (green boxes) are replaced. The representative examples are highlighted in red boxes.

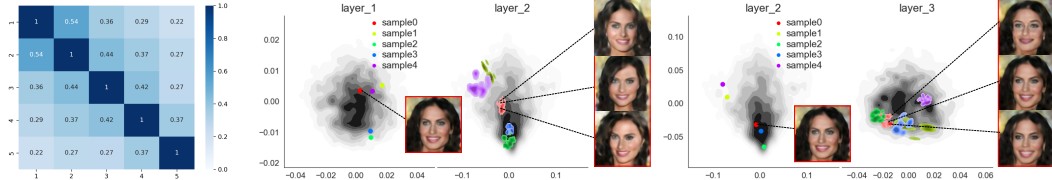

Figure 7: Layer-wise hierarchy analysis. (a): correlation between cross-layer latents, obtained on the train-split of CelebA-HQ at Stage-1. (b) and (c): visualization of conditional distributions across layers. The gray regions show the distribution of latents from Stage-1, while the colored regions represent the sampled latents from Stage-2.

as facial orientation, remain unchanged. In the last row, only skin tone changes. More examples can be found in the Appendix.

It's important to note that latent composition disrupts the conditional chain, meaning the newly composed latents may be incompatible across layers, resulting in lower-quality images. Nevertheless, we conduct latent composition to illustrate how and why our method works. The results show that the image semantics can be embedded and disentangled in parameter space, offering a new perspective on image generation.

### 4.3.2 LAYER-WISE HIERARCHY ANALYSIS

**Layer-wise correlation.** We perform layer-wise correlation analysis on the latents to show the necessity of conditional dependency modeling. We compute the cross-layer correlation between latents using Singular Vector Canonical Correlation Analysis (SVCCA) (Raghu et al., 2017), a metric that measures the correlation between neural network representations. Fig. 7 (a) displays the pairwise correlations between $\mathbf{h}$ across layers, trained on Celeba-HQ in Stage-1, showing the non-negligible correlation between layers. This underscores the importance of modeling conditional distributions $p(\mathbf{h}^l|\mathbf{h}^{<l})$, rather than independent marginal distributions $p(\mathbf{h}^l)$ in Stage-2. More results on other datasets can be found in the Appendix. We next show the learned hierarchical structures of latents.

**Layer-wise dependency visualization.** We visualize the hierarchical dependencies of latents to better understand how data semantics are encoded in INR's weight space, as the latents are mapped to gating vectors that modulate the experts at each layer. Fig. 7 (b) and (c) show the conditional distributions of latents from adjacent layers, trained on CelebA-HQ. Specifically, we apply PCA to the latents and plot their distributions in grayscale. For example, the Fig. 7 (b) shows distributions for layers 1 and 2. We then run the generation process five times to obtain five sampled latents at each layer, as depicted in color in the left parts of Fig. 7 (b) and (c). Based on the HCDM, we can plot the resulting conditional distributions of latents at each subsequent layer, represented by colored regions in the right parts. Additionally, we show the final generated images corresponding to different samples. We can see clear patterns of a hierarchical structure, which corresponds to the semantic variations at different granularities. For example, the sampled latents in layer 1 determine the overall contours of the face. When layer 2 is determined, variations in layer 3 modify the facial expressions while keeping the orientation fixed. In fact, different layers govern different aspects of

the generated data, as we have seen in the compositional analysis. Furthermore, the latent sampling space at each layer is constrained by the preceding layers, ensuring compatibility between layers in representing the data. Therefore, the generated data semantics are hierarchically controllable.

### 4.4 ABLATION STUDIES

**Ablation on condition modeling.** To demonstrate the importance of conditional dependency modeling for hierarchical controllable generation, we train an unconditional diffusion model that directly maps noise to layer-wise latents in Stage-2. We then sample $p(\mathbf{h}^l)$ independently for $l = 1, \cdots, L$ to generate the full latents. The resulting images for CelebA-HQ are shown in Fig. 8 (a), while images generated with conditional modeling are shown in (b) for comparison. Although human faces are recognizable in (a), the noticeable artifacts highlight

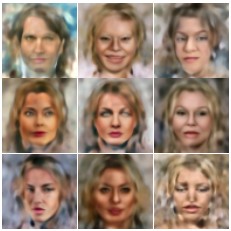 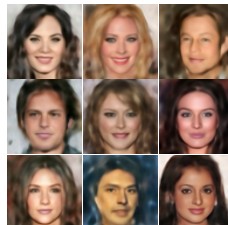

(a) Without condition      (b) With condition

Figure 8: Ablations on conditional modeling.

that independently sampled layer-wise latents fail to ensure consistent semantic composition across layers. In contrast, conditional modeling successfully achieves this compatibility.

Table 2: Ablation study of different binary condition lengths, 8, 12, 15, and 20, when training Stage-2 on CelebA-HQ, ShapeNet, and SRN-Cars datasets.

| Model | CelebA-HQ | | | | | ShapeNet | | | | | SRN-Cars | | | | |
|---|---|---|---|---|---|---|---|---|---|---|---|---|---|---|---|
| | $std_1$ | $std_{12}$ | $std_{23}$ | $std_{34}$ | $std_{45}$ | $std_1$ | $std_{12}$ | $std_{23}$ | $std_{34}$ | $std_{45}$ | $std_1$ | $std_{12}$ | $std_{23}$ | $std_{34}$ | $std_{45}$ |
| $HCDM_8$ | 0.7766 | 0.9153 | 0.9363 | 0.8772 | 0.8631 | 0.8533 | 0.9510 | 0.8933 | 0.7682 | 0.7301 | 0.7223 | 1.1270 | 1.1108 | 0.9474 | 1.1210 |
| $HCDM_{12}$ | 0.7766 | 0.9092 | 0.9292 | 0.8663 | 0.8550 | 0.8533 | 0.9424 | 0.8712 | 0.7327 | 0.7110 | 0.7223 | 1.0663 | 1.0180 | 0.9331 | 1.1803 |
| $HCDM_{15}$ | 0.7766 | 0.5344 | 0.5478 | 0.3257 | 0.2570 | 0.8533 | 0.6125 | 0.5241 | 0.4088 | 0.3857 | 0.7223 | 0.6348 | 0.6432 | 0.5837 | 0.6578 |
| $HCDM_{20}$ | 0.7766 | 0.1032 | 0.1121 | 0.0853 | 0.0766 | 0.8533 | 0.1051 | 0.0823 | 0.0715 | 0.0522 | 0.7223 | 0.1048 | 0.1122 | 0.0821 | 0.0933 |

**Ablation on binary condition.** We demonstrate that the length of binary conditions impacts the effectiveness of learning conditional dependencies, as shown in Table 2. We set the binary lengths to $8, 12, 15, 20$ and train the HCDM on different datasets. Initially, we sample 5000 latents and compute the standard deviation of the first part, denoted as $\mathbf{std}_1$. Since the first part has no conditions and is sampled from noises, it shows high values irrelevant to the binary lengths. Subsequently, we select 10 random samples from the first part to use as conditions and get 5000 samples for the second part. Here, the standard deviation, denoted as $\mathbf{std}_2$, decreases as the binary condition length increases, because longer binary conditions contain more information from preceding layers. Once the length reaches a certain threshold, the standard deviation approaches zero, turning the conditional chain into a direct one-to-one mapping, thus diminishing controllability. However, if the length becomes too small, e.g. 0, all parts will be independent thus losing conditional dependency. Therefore, we empirically set the length to be 12 for Stage-2 training. We repeat this procedure for other parts and observe similar results.

## 5 CONCLUSION

In this work, we proposed the Controllable Hierarchical Implicit Neural Representation (CHINR) framework, addressing the limitations of existing generative INRs that learn joint parameter distributions while ignoring the hierarchical structure of parameters and data semantics. By structuring the INR as a Layers-of-Experts (LoE) network and leveraging a Hierarchical Controllable Diffusion Model (HCDM), our approach captures conditional dependencies across layers, improving generalizability and enabling controllable data generation.

One limitation is scalability to larger datasets, as the shared network may struggle to capture complex and diverse data patterns. A possible solution is to incorporate local information using patchwise modulation (Mehta et al., 2021; Bauer et al., 2023) or INRs with localized nonlinearity (e.g., WIRE (Saragadam et al., 2023)). Future directions include exploring sparse gatings, as in Mixture of Experts (MoE) methods Wang et al. (2022), to promote expert diversity and specialization. Additionally, the learning of layer-wise semantic hierarchy in Stage-1 can be guided through predefined attributes or deep clustering. This would allow the model to develop more interpretable and distinct semantics across layers, improving control over fine-grained details or desired characteristics.

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

## A    DETAILS ON EXPERIMENTAL SETUP

**Implementation details.**   The LoE structure can be configured with the number of layers $L$, the number of experts at each layer $K$, the channel dimension of each expert $C$, and the dimension of the latent at each layer $H$, denoted as a tuple $(L, K, C, H)$. We train LoEs of $(7, 384, 128, 128)$, $(5, 256, 64, 256)$, $(6, 256, 64, 64)$, and $(5, 64, 64, 64)$ in CelebA-HQ (Karras, 2017), ShapeNet (Chang et al., 2015), SRN-Cars (Sitzmann et al., 2019), and AMASS (Mahmood et al., 2019) datasets, respectively. We follow mNIF (You et al., 2024) on the data processing protocols for CelebA-HQ, ShapeNet, and SRN-Cars datasets. Details about the AMASS dataset are provided in Sec. B.3.

**Training details.**   In Stage-1, we train LoEs via meta-learning on CelebA-HQ, ShapeNet, and AMASS, and with auto-decoding on SRN-Cars. We use a batch size of 32, an outer learning rate of $1e-4$, an inner learning rate of 1 with 3 steps, and train the LoE for 800 epochs in the meta-learning setting. For auto-decoding experiments on SRN-Cars, we use a batch size of 8, a learning rate of $1e-4$, and train the LoE for 1000 epochs. In both settings, we use the AdamW (Loshchilov, 2017) optimizer without weight decay. In Stage-2, we set the training batch size to be 32, learning rate $1e-4$, and cosine scheduler with minimum learning rate 0.0. We train the HCDM for 1000 epochs with the AdamW optimizer.

## B    ADDITIONAL EXPERIMENTAL RESULTS

### B.1    GENERALIZABILITY ANALYSIS THROUGH RETRIEVAL

Generated                    Retrieved from training set

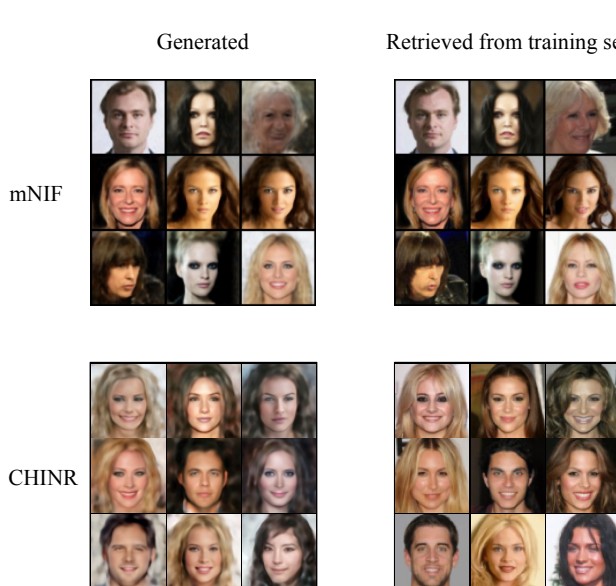

Figure 9: Retrieval on CelebA-HQ: mNIF retrieves images closely resembling those from the training set, while CHINR demonstrates better generalization by producing distinct new images.

We use retrieval to compare the generalizability of CHINR and mNIF on the CelebA-HQ dataset. Specifically, we generate samples and retrieve the closest images from the training set. As shown in Fig. 9, mNIF generates samples that are very similar to the training images, suggesting a higher chance of "memorization". In contrast, CHINR demonstrates better generalization by producing "new" samples that differ more noticeably from the training data.

### B.2    MORE GENERATED SAMPLES

Fig. 10 shows more generated samples on CelebA-Net, ShapeNet, and SRN-Cars datasets.

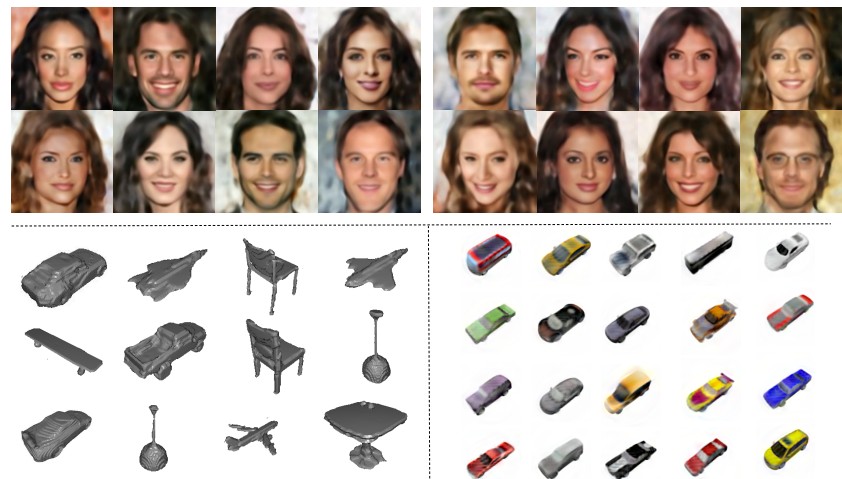

Figure 10: More generated samples of CelebA-HQ, ShapeNet, and SRN-Cars data.

Table 3: Quantitative results on AMASS.

| Model | MSE↓ |
|---|---|
| mNIF (You et al., 2024) | 0.015 |
| CHINR | 0.011 |

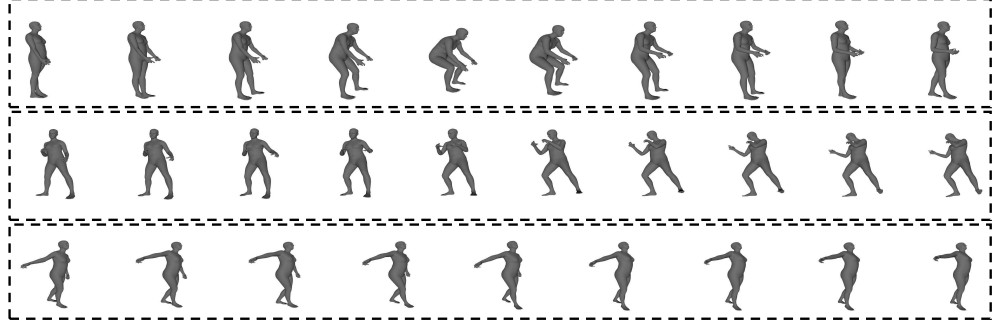

Figure 11: Generated motions with HCDM. each row denotes one sampled data.

### B.3 AMASS EXPERIMENTS

We apply our proposed CHINR model to the AMASS dataset of 3D human motions. For each motion sequence, we use 200 frames, with each frame represented by 165 values corresponding to the locations and rotations of body joints. As a result, each data instance is formatted as a grid with size $200 \times 165$. In Stage-1, the LoE is employed to fit the motion instances. In Stage-2, we set the binary lengths to 8 to avoid memorizing conditions.

**Reconstruction and generation results.** The reconstruction performance is shown in Table. 3. The randomly generated motions are shown in Fig. 11.

**Semantic-level Interpolation.** Since the LoE successfully learns the consistent latent space, we can perform semantic-level interpolation for motions. As shown in Fig. 12, given two fitted sequential motions with LoE, each corresponds to a latent, we can interpolate the latent from the start motion (indicated by the red dashed box) to the end motion (indicated by the green dashed box) linearly with ratio $[0.2, 0.4, 0.6, 0.8]$. We can see that the interpolated motions change smoothly from the

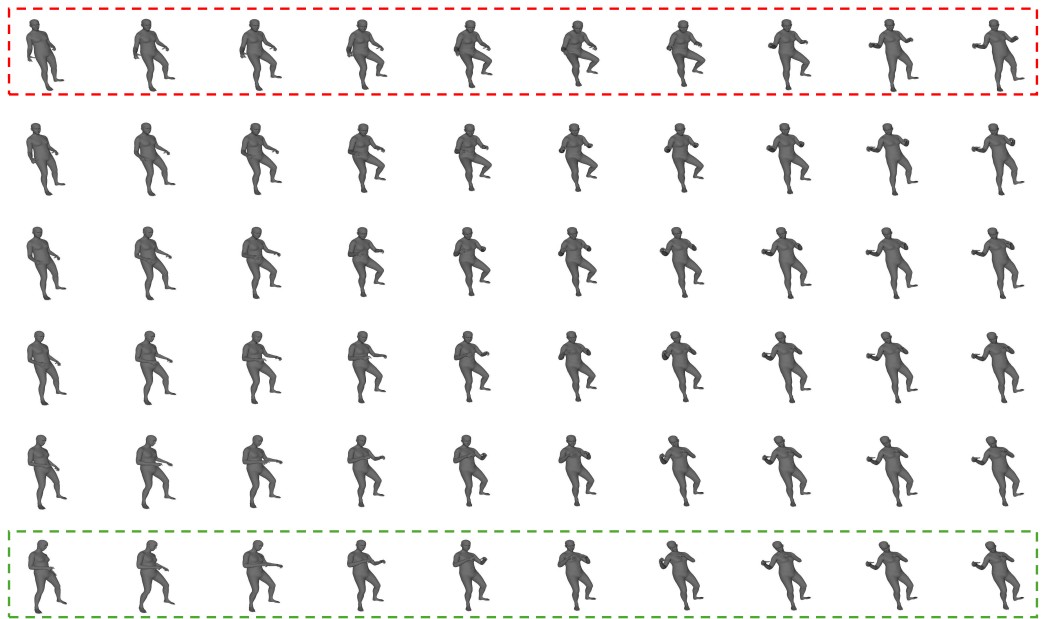

Figure 12: Semantic interpolation for AMASS data. Anchor sequential motions (indicated by the red and green dashed boxes) are first fitted with LoE to obtain latents. Then semantic-level interpolation is performed by interpolating the latents. The red dashed box denotes the start motion, and the green dashed box denotes the end motion.

start to the end. Semantic-level interpolation can be useful in the gaming industry, and 3D-digital content generation.

**Temporal-level interpolation.** Since the INR can generate data instances in any resolution, we can easily enlarge the input coordinates' resolution in the time dimension to achieve temporal-level interpolation. We set the length of the time dimension to be 200 and 400, then get motions with LoE. The interpolated results are submitted as videos named "motion_short.mp4" and "motion_long.mp4".

### B.4 HIERARCHICAL CONTROLLABLE GENERATION

More examples of hierarchical controllable data generation are presented in Fig 13.

### B.5 LATENT-BASED RETRIEVAL

We show an application of data retrieval by latents, since they already embed rich semantic meanings. We first obtain the latents for the target data by fitting it to the LoE through a few gradient steps. Once the latents are optimized, they can be used to retrieve similar data by comparing their latent representations to the searched set, allowing us to search for semantically similar examples within the latent space. Fig. 14 shows this process by using images from the test-split of CelebA-HQ as the targets, and train-split images as the searched set. We demonstrate two approaches for retrieval: (1) using the flattened $\mathbf{h}$ for all layers, and (2) layer-wise retrieval using each layer's latent $\mathbf{h}^l$. As shown in Fig. 14, retrieval by the flattened $\mathbf{h}$ will retrieve samples that are broadly similar, while layer-wise retrieval retrieves samples with specific semantic similarities. For example, $\mathbf{h}^2$ retrieves faces with similar orientations, while $\mathbf{h}^3$ retrieves faces with similar facial features such as eye shape and expressions.

## C ANALYSIS

In this section, we provide more analysis of the latent space and the functionalities of binary conditions.

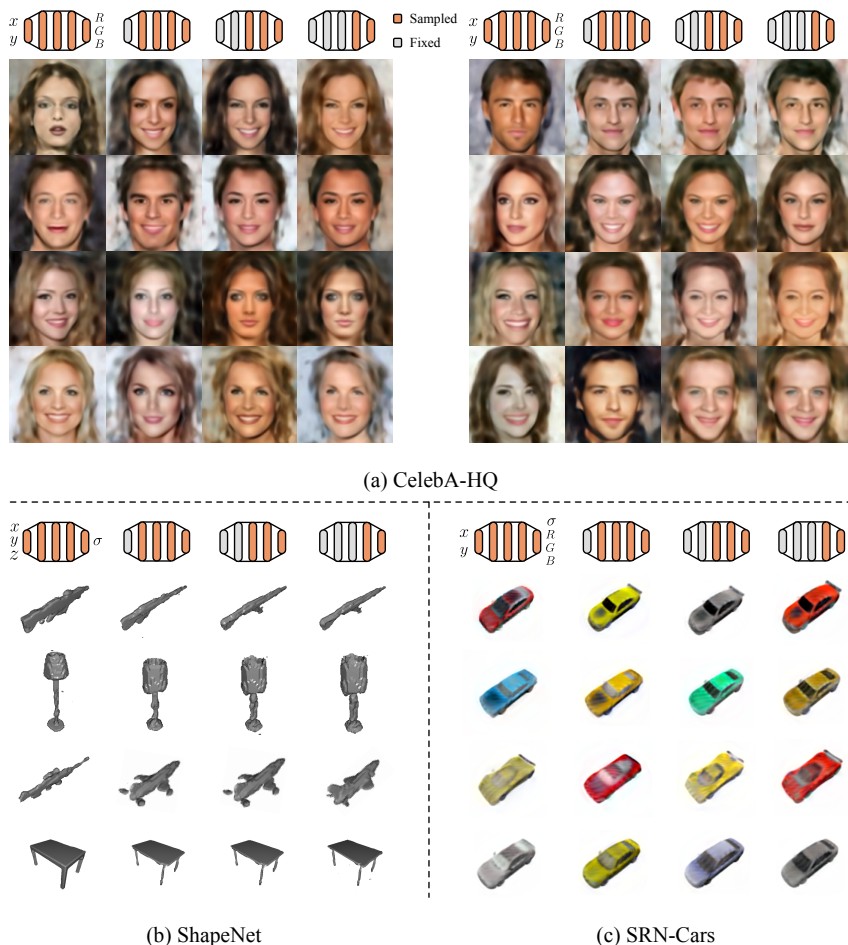

(a) CelebA-HQ

(b) ShapeNet

(c) SRN-Cars

Figure 13: More examples of hierarchical controllable generation on CelebA-HQ, ShapeNet, and SRN-Cars data.

## C.1 LATENT SPACE ANALYSIS

Here, we analyze the latent space further, focusing on its interpolation capabilities and providing additional results of correlation analysis.

### C.1.1 LATENT INTERPOLATION

To illustrate that our model learns a consistent and metric latent space, following definitions in Du et al. (2021), we perform latent space interpolation in two ways: complete interpolation, and layer-wise interpolation.

**Complete Interpolation** is shown in Fig.15. Four corners present the signals with latent generated from Stage-1. The intermediary signals are bilinearly interpolated from four corners in latent space. The results demonstrate that the learned latent is metric and consistent with human perception.

**Layer-wise Interpolation.** Since our LoE embeds semantics hierarchically in different parts of the latent, we can interpolate each part to control specific semantics. As shown in Fig. 16, we interpolate the second, third, and fourth parts of the latent associated with red-boxed signals, with the corresponding parts of the right side latent. For CelebA-HQ samples, we find that the facial orientation, facial features, and skin tone can be interpolated independently. This demonstrates that each part of the latent also constructs a metric and consistent manifold.

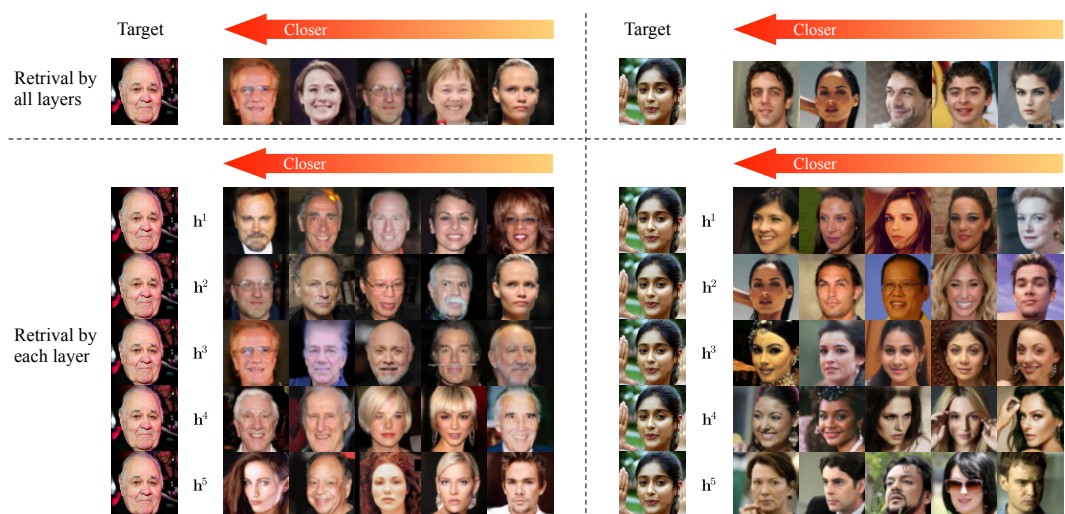

Figure 14: Latent-based retrieval via two approaches: retrieval by all layers and retrieval by each layer.

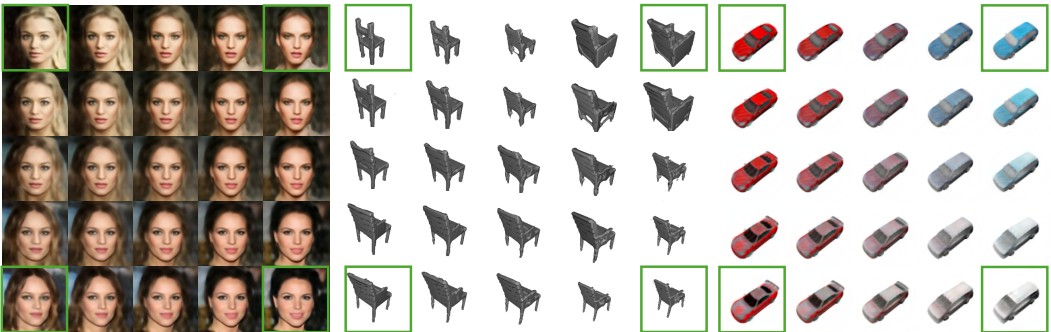

Figure 15: Latent space interpolation is performed for LoE, with four corner points representing the anchor examples rendered in stage 1. The intermediary points are generated through the bilinear interpolation of the latents associated with these four anchors. The interpolation is evaluated on datasets CelebA-HQ, ShapeNet, and SRN-Cars.

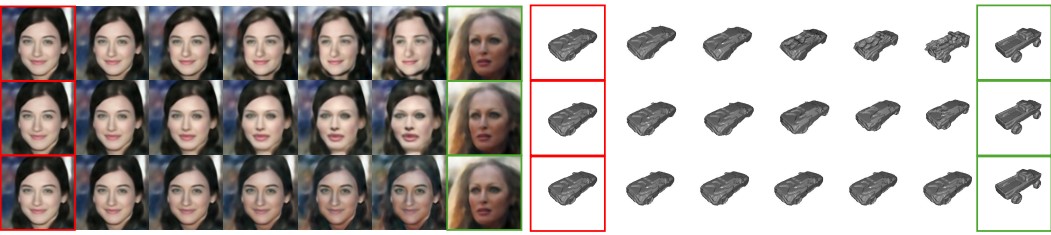

Figure 16: Layerwise interpolation. The red boxes denote the start and the green boxes denote the end. For the CelebA-HQ, the layers $2 \rightarrow 4$ are interpolated respectively while other layers are fixed. For the ShapeNet, the layers $1 \rightarrow 3$ are interpolated respectively.

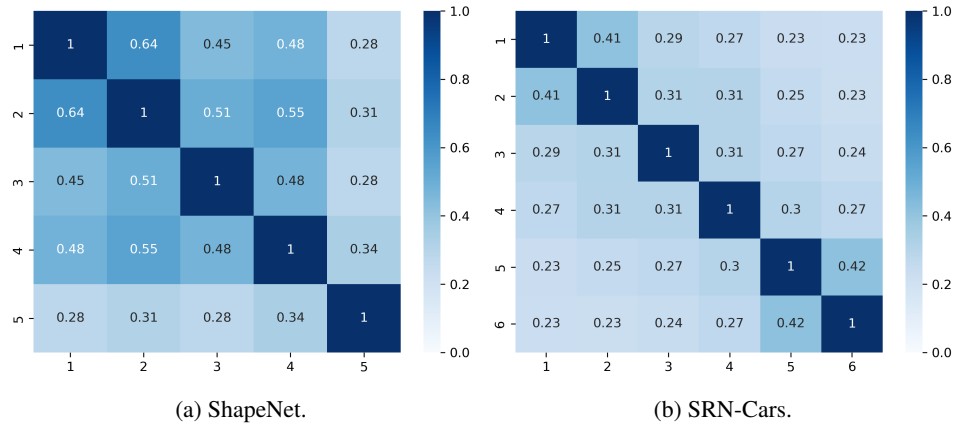

(a) ShapeNet.                              (b) SRN-Cars.

Figure 17: Correlation between the learned latents across layers, trained on ShapeNet (Chang et al., 2015) and SRN-Cars (Sitzmann et al., 2019). The non-negligible correlation between adjacent layers (e.g., $\mathbf{h}^1$ and $\mathbf{h}^2$) reveals the necessity of conditional distribution learning.

### C.1.2 LAYER-WISE CORRELATION ANALYSIS

We provide correlation analysis on additional datasets in Fig. 17 and Fig. 18. This highlights the significance of conditional modeling in the hierarchical generation process.

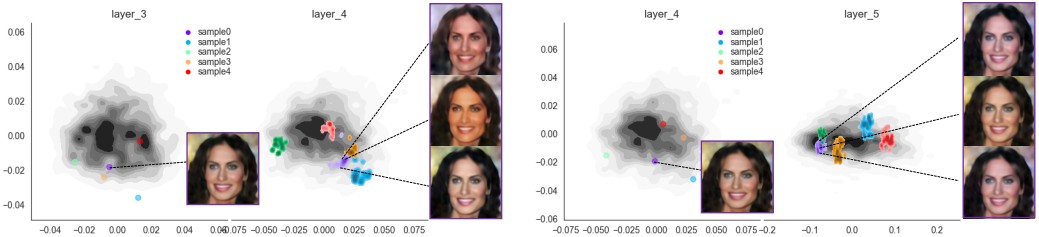

Figure 18: Visualization of conditional distributions across layers $3, 4, 5$. The gray regions present the distribution of latents from Stage-1, while the colored regions represent the sampled latents from Stage-2.

### C.2 BINARY CONDITION ANALYSIS

We analyze the clustering of latents and binary conditions on CelebA-HQ dataset, as shown in Fig. 19. Firstly, we use the KMeans algorithm to get 10 clusters of latents, shown as the dots in the figure. Then we select three anchor latents, generate three binary conditions with HCDM, and search the nearest binary-corresponded latents. The nearest neighbors are represented by the stars. We can observe that the binary conditions embed the latents' information and form a consistent binary condition space. This binary condition space corresponds to the latent space.

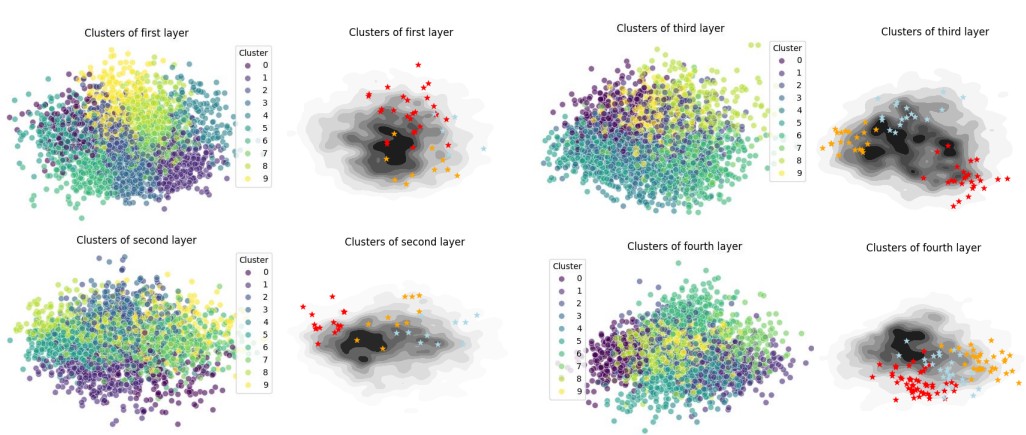

Figure 19: Clusters of each part of latent and binary conditions. The dotted plot presents clusters of each part of latents trained on ClebA-HQ. The gray distribution plot presents the distribution of each part of latents, and starred scatter plot presents clusters of latents with similar binary conditions.

