# OpenReview forum: "Controllable Data Generation with Hierarchical Neural Representations"
_ICLR.cc/2025/Conference — Submitted to ICLR 2025_

### Official Review · Reviewer_ww4M · 2024-10-31

**Soundness:** 3
**Presentation:** 3
**Contribution:** 3
**Rating:** 6
**Confidence:** 3

**Summary:**

The paper introduces a novel framework called Controllable Hierarchical Implicit Neural Representation (CHINR), aimed at enhancing data generation and control through hierarchical neural representations. CHINR leverages implicit neural representations (INRs) to represent complex data as continuous functions and incorporates hierarchical dependencies across layers to improve generalizability and control. The framework operates in two stages: first, it builds a Layers-of-Experts (LoE) network where each layer uses specific latent vectors to manage distinct data features, allowing for a disentangled representation. In the second stage, a Hierarchical Controllable Diffusion Model (HCDM) is used to capture conditional dependencies across layers, enabling nuanced control of generated content. Experiments across various datasets (e.g., CelebA-HQ, ShapeNet, and AMASS) show that CHINR offers improved generalizability and hierarchical control over generated data.

**Strengths:**

1. CHINR’s two-stage framework with LoE and HCDM effectively captures hierarchical structures in data, enhancing representation control and generalisability.
2. The model is rigorously evaluated across multiple modalities, showing consistent improvement over baseline methods, especially in quality and generalisation.
3. The model’s ability to control data generation at multiple granular levels (e.g., overall structure, fine details) is a substantial advancement for applications requiring precision.

**Weaknesses:**

see questions.

**Questions:**

1. Could the proposed binary conditions be replaced or optimised further to improve interpretability or training efficiency?
2. How does CHINR handle data types with less hierarchical structure (e.g., text), and is its control mechanism adaptable to such cases?
3. How sensitive is the hierarchical control to variations in input data quality, and could this affect generalisation in real-world applications?

---

> ### Author Response · Authors · 2024-11-22
> **Response to Reviewer ww4M (1/3)**
>
> We appreciate valuable comments from Reviewer ww4M.
> # Q1 Optimize binary conditions
> 1. Yes, a promising approach for **interpretability** enhancement is to establish correspondences between binary conditions and ground-truth class labels. Incorporating these class labels during training enables control based on class labels or textual inputs.
>
> 2. The binary condition has no apparent effect on training **efficiency**.

---

> ### Author Response · Authors · 2024-11-22
> **Response to Reviewer ww4M (2/3)**
>
> # Q2 CHINR for unstructured data
>
> Our method can **represent** any structured signal as a function, i.e., a mapping from input coordinates to output values. To represent unstructured data such as texts, **we can first project unstructured texts into structured embedding space** with dimension $L$, so a phrase with $N$ words can be represented as a function: $f:x\rightarrow T: \in \mathbb{R}^{N}\rightarrow\mathbb{R}^{N\times L}$. Then the CHINR can be trained to represent any phrase with positions $x \in [1\cdots N]$ as input, and $T$ as output.
>
> However, **the control mechanism will not work on text embeddings**, because the textual embedding space lacks an inherent hierarchical structure that aligns with spatial frequency, i.e. the low-frequency components represent the overall structure while high-frequency components represent finer details.
>
> Based on the analysis, a promising future for hierarchical control of textual data is to **figure out an embedding space that has a hierarchical structure aligned with spatial frequency,** then the CHINR will work.

---

> ### Author Response · Authors · 2024-11-22
> **Response to Reviewer ww4M (3/3)**
>
> # Q3 Sensitivity to quality variations
>
> The sensitivity of hierarchical control to quality variations is reflected in two aspects:
>
> 1. The reconstruction quality in stage 1 caps the controllability. If the proposed LOE cannot learn hierarchically structured latents in Stage 1, then the hierarchical control facilitated by generation in Stage 2 is inhibited. We find that when the PSNR on CelebA-HQ reaches above 25, there won't be visually apparent distortions, the hierarchical control works well.
>
> 2. The quality of ground-truth data for stage 1 caps the controllability. That is because the larger noise level makes learning conditional dependency harder. In an extreme case where the data consists solely of noise, the learned latents will lose any meaningful dependencies. To demonstrate the analysis, we experiment by perturbing ground-truth data with various noise levels as shown in the table below. As the noise level in ground-truth data increases, the quality of both reconstruction and generation deteriorates. The range of RGB values for ground-truth data is $[0,1]$.
>
> | **Noise-level**           |  PSNR↑ | SSIM↑    | FID↓ |
> |----------------------------|--------------------------|----------|----------------------|
> | $+0.0*\mathcal{N}(0,1)$    | 34.9                    | 0.963    | 13.4                |
> | $+0.1*\mathcal{N}(0,1)$    | 29.2                    | 0.905    | 18.2                |
> | $+0.3*\mathcal{N}(0,1)$    | 22.7                    | 0.732    | 55.3                |
> | $+0.6*\mathcal{N}(0,1)$    | 17.3                    | 0.421    | 158.5               |
>
> *Table*: Sensitivity of conditionally sampled data to noise in ground-truth data, evaluated on CelebA-HQ.
>
> Therefore, in real-world applications, it should be ensured the data has no apparent distortions.

---

> ### Author Response · Authors · 2024-11-22
>
> Thanks for your insightful questions. We hope our answers address your concerns. Please let us know if you have any further questions.

---

> > ### Comment · Reviewer_ww4M · 2024-11-23
> > **reply**
> >
> > Thanks for the author's detailed responses. Considering the overall contribution, I would like to keep my score.

---

> > > ### Author Response · Authors · 2024-11-24
> > > **Response to Reviewer ww4M**
> > >
> > > Thank you for recognizing our contributions. We sincerely apperciate your efforts and will ensure to consider and incorporate any future suggestions.

---

### Official Review · Reviewer_LwTN · 2024-11-01

**Soundness:** 3
**Presentation:** 2
**Contribution:** 2
**Rating:** 6
**Confidence:** 3

**Summary:**

The authors exploit a hierarchical and sparse structure prevalent in INRs to conceive a data generation method that is more controllable and modular than contemporary approaches.

I am far from knowledgable in this field. Take my review with a heap of salt.

**Strengths:**

- The compositionality aspect of this method is enticing
- The controllability and clear hierarchical order (Fig 1) is fascinating
- The premise of utilizing the hierarchical structure of the SIREN INR seems intuitive and reasonable to me.

**Weaknesses:**

- Table 1 needs uncertainties to gauge superiority or consistency. The method  does not seem super strong on the performance front.
- Given the discrepancy in the conditional chain (and thus worse image quality) when controlling the latents as done in Fig. 6 (see L456ff), is the controllability aspect still a valid bonus? Is this a drawback that can be retroactively fixed by somehow iterating multiple times over the hierarchies?

**Questions:**

- How does it happen that, for instance in the middle of Figure 1, the hierarchies so nicely align with human abstractions over the data? It seems somewhat coincidental to me that the first layer corresponds to the type of object (chair, jet, table) and the next layer corresponds to the type of chair, and so on.
- It would be nice to see an equation for the LoE. Is each expert a SIREN model? Are the experts fully connected, and then controlled by a layer-specific gating function?
- In eq.3, isn’t $p(h^l|h^{<l}) = p(h^l|h^{l-1})$ (i.e. markov)?
- The design of the condition seems very ad-hoc, specifically the quantization. Can you motivate this further beyond “removing enough information to not overtrain”? Since the very coarse quantization causes a discontinuity in the condition, I expect this to have weird side effects, albeit maybe not super prominent in this domain.
- Nit: citation needed in L32 for INRs

---

> ### Author Response · Authors · 2024-11-22
> **Response to Reviewer LwTN (1/6)**
>
> We appreciate the insightful feedback from Reviewer LwTN.
> # W1 Table 1 needs uncertainties
>
> ## Uncertainty
>
> Thanks for pointing out this. Table 1 presents results from published papers to ensure fair comparisons, though none of the methods report uncertainty metrics. We understand your concern and retrain these methods four times under identical settings but with different random seeds. The uncertainties, i.e. standard deviation, are reported in tables below.
>
> The results for CelebA-HQ are reported in:
>
> | **Model** | PSNR↑ | SSIM↑         | FID↓ |
> |-----------|---------------------------|---------------|----------------------|
> | Functa [1]    | 26.2 ± 0.3               | 0.795 ± 0.015 | 41.0 ± 0.2          |
> | GEM [2]      | 26.5 ± 0.4               | 0.814 ± 0.018 | 30.8 ± 0.3          |
> | GASP [3]     | 31.6 ± 0.8               | 0.902 ± 0.021 | 13.6 ± 0.3          |
> | mNIF [4]     | 34.5 ± 0.2               | 0.957 ± 0.005 | **13.2 ± 0.1**          |
> | CHINR     |  **34.9 ± 0.3**       |  **0.964 ± 0.006**| 13.4 ± 0.1        |
>
> The results for ShapeNet are reported in:
>
> | **Model** |  PSNR↑ | Accuracy↑      |  Coverage↑ | MMD↓             |
> |-----------|---------------------------|----------------|---------------------------|------------------|
> | Functa    | 22.1 ± 0.3               | 0.983 ± 0.005 | 0.437 ± 0.005            | 0.0013 ± 0.0002 |
> | GEM       | 21.4 ± 0.4               | 0.977 ± 0.007 | 0.408 ± 0.003            | 0.0016 ± 0.0003 |
> | GASP      | 16.7 ± 0.8               | 0.928 ± 0.011 | 0.343 ± 0.009            | 0.0023 ± 0.0006 |
> | mNIF      | 21.4 ± 0.3               | 0.975 ± 0.008 | 0.435 ± 0.003            | 0.0014 ± 0.0003 |
> | CHINR     | **22.3 ± 0.2**               | **0.988 ± 0.005** | **0.441 ± 0.002**            | **0.0011 ± 0.0001** |
>
> The results for SRN-Cars are reported in:
>
> | **Model** |  PSNR↑ | SSIM↑         |  FID↓ |
> |-----------|---------------------------|---------------|----------------------|
> | Functa    | 24.3 ± 0.2               | 0.738 ± 0.009 | 80.1 ± 0.2          |
> | mNIF      | 26.0 ± 0.3               | 0.763 ± 0.013 | 79.3 ± 0.3          |
> | CHINR     | **26.3 ± 0.2**               | **0.780 ± 0.011** | **77.8 ± 0.2**          |
>
> ## Strengths
> While Table 1 demonstrates competitive generation performance, our primary focus is on controllable generation through hierarchical modeling—key aspects not addressed by prior work.
> Overall, our contributions are highlighted in four aspects: **controllability, generalizability, versatility**, and **quality**.
>
> - **Controllability**: Our work distinguishes from previous methods in the controllability of generating fine-grain details, which has never been explored in the implicit neural representation (INR) community.
>
> - **Generalizability**: Our work can avoid trivial memorization thus generating more diverse data compared with the SOTA methods, such as mNIF. Please find the discussion and visual samples from Line 357-363 in the main paper and Figure 9 in the supplementary material, respectively.
>
> - **Versatility**: The CHINR's performance is validated across four different data domains, i.e. facial images, point clouds, NeRFs, and motions. This demonstrates the proposed hierarchical control and conditional dependency modeling is broadly applicable to data semantics with an inherent hierarchy structure.
>
> - **Quality**: Our work outperforms the SOTA methods on both the reconstruction and generation metrics, which can be seen from Table 1 in the main paper.
>
> [1] Emilien Dupont, Hyunjik Kim, SM Eslami, Danilo Rezende, and Dan Rosenbaum. From data to functa: Your data point is a function and you can treat it like one, ICML 2022.
>
> [2] Yilun Du, Katie Collins, Josh Tenenbaum, and Vincent Sitzmann. Learning signal-agnostic manifolds of neural fields, NeurIPS 2021.
>
> [3] Emilien Dupont, Yee Whye Teh, and Arnaud Doucet. Generative models as distributions of functions, AISTATS 2022.
>
> [4] Tackgeun You, Mijeong Kim, Jungtaek Kim, and Bohyung Han. Generative neural fields by mixtures of neural implicit functions, NeurIPS 2024.

---

> ### Author Response · Authors · 2024-11-22
> **Response to Reviewer LwTN (2/6)**
>
> # W2 Discrepancy in the conditional chain affects quality
> Thank you for your insightful feedback. Controllability is a valid and valuable bonus ensured by sampling in a conditional chain. However, **Figure 6 does not show the conditional chain**. It is a latent replacement experiment to reveal that the layer-wise latents embed distinct and compositional semantics. As expected, the replacement disrupts the conditional chain, affecting image quality. In contrast, our controllable generation follows the conditional chain to ensure compatible latents for better quality.
>
> We acknowledge that the conditional chain can pose challenges for generation quality due to potential error accumulation. For instance, if a generated latent lies in a low-density region of the learned distribution, subsequent layers may struggle to produce meaningful and compatible outputs. However, **iterating multiple times, being time-consuming, does not essentially guarantee sampling in high-density regions.** Instead, we address this in both training and inference of the conditional diffusion models.
>
> 1. During training, we first train the model to learn inter-layer dependency, using ground truth layer-wise latents (from Stage 1) to form conditions. Then, we fine-tune the model to learn a reasonable condition chain by gradually using generated latents to form conditions.
>
> 2. During inference at each layer, we further mitigate issues by adjusting the standard deviation of input noise for the diffusion process, reducing the likelihood of generating outlier latents that degrade quality.

---

> ### Author Response · Authors · 2024-11-22
> **Response to Reviewer LwTN (3/6)**
>
> # Q1 Hierarchy alignment
>
> Thank you for your insightful question. The alignment of hierarchies in generations with human abstractions, as shown in Figure 1, is supported by: 1. the hierarchical representation ability of INRs (Sec. 2.2), embedding coarse-to-fine semantics across layers, and 2. the conditional chain in HCDM, ensuring their coherent composition. Specifically,
>
> 1. INRs naturally exhibit a spectral bias, where early layers capture coarse, low-frequency components (e.g., object categories like chair, jet, table), and later layers refine these with higher-frequency details (e.g., types of chairs). This inherent characteristic of INRs underlies the observed alignment with human abstractions.
>
> 2. In our method, the conditional chain ensures that coarse structures (e.g., object categories) are determined first, and the later layers are constrained to finer details (e.g., subcategories or textures). This controlled generation process reflects the hierarchical nature of the data.
>
> Therefore, by embedding then conditional sampling, these two factors collaboratively generate human abstraction-aligned data as shown in Figure 1.

---

> ### Author Response · Authors · 2024-11-22
> **Response to Reviewer LwTN (4/6)**
>
> # Q2 The equation for the LoE
>
> Thank you for your suggestion. We agree that providing an equation for the Layers-of-Experts (LoE) structure would improve clarity. **Each expert in the LoE is a fully connected layer, not a SIREN model**. So the parameters of expert $k$ at layer $l$, denoted by $\boldsymbol\theta^l_k$ as appeared in Sec. 3.1, contain a weight matrix and a bias vector. **The experts are controlled by a layer-specific gating function**, which determines their contributions at each layer. Nonlinear activation (sine) is applied after the experts are combined. The equation for the LoE INR at layer $l$ is
> \begin{equation}
> \begin{split}
>     \boldsymbol y^{l+1} = \sin(\omega_0\cdot(\bar{\boldsymbol\theta}^l\cdot\boldsymbol y^l)),~~
>     \bar{\boldsymbol\theta}^l&=\sum_{k=1}^K\boldsymbol\theta_k^l\cdot \alpha_k^l,\\\\
>     [\alpha_1^l,\alpha_2^l,\cdots,\alpha_K^l]^\top=\boldsymbol\alpha^l&=g_{\boldsymbol\phi}(\mathbf h^l),
> \end{split}
> \end{equation}
> where $\boldsymbol y$ represents output of each layer, $\omega_0$ is a constant factor, and $\bar{\boldsymbol\theta}^l$ denotes parameters at layer $l$, modulated by a *gating vector* $\boldsymbol\alpha^l$, which is computed by the gating module $g_{\boldsymbol\phi}(\cdot)$. $\mathbf h^l$ denotes the $l_{th}$ layer of latent $\mathbf h$.
>
> We have included this equation in the revised paper.

---

> ### Author Response · Authors · 2024-11-22
> **Response to Reviewer LwTN (5/6)**
>
> # Q3 Markov in eq.3
>
> Thanks for your question. In Equation 3 the expression is $p(\mathbf h^l|\mathbf h^{<l})$ rather than $p(\mathbf h^l|\mathbf h^{l-1})$ (as in Markov chain). It is noted that $p(\mathbf h^l|\mathbf h^{<l})$ represents the dependency in layers of latents, which is not the Markov in backward diffusion steps.
>
> **Mathematically**, $p(\mathbf h^l|\mathbf h^{<l})$ arises from the factorization of joint distribution into marginalized and conditional distributions. For example, for $\mathbf h=[\mathbf h^1, \mathbf h^2, \mathbf h^3]$, the joint distribution $p(\mathbf h)$ can be written by the chain rule as:
> \begin{align}
> p(\mathbf h) &= p(\mathbf h^1)p(\mathbf h^2|\mathbf h^1)p(\mathbf h^3|\mathbf h^2,\mathbf h^1)\\\\
> &= p(\mathbf h^1)\prod^{3}_{l=2}p(\mathbf h^l|\mathbf h^{<l}).
> \end{align}
>
> **Intuitively**, this reflects the hierarchical nature of the layer-wise latents. Consider a hierarchy of facial attributes (orientation--expressions--shape of the eyes), the fine details of the eyes depend on both the orientation and expressions, not just on the immediate predecessor. This fundamentally differs from a Markov chain, where each state depends only on the previous state.

---

> ### Author Response · Authors · 2024-11-22
> **Response to Reviewer LwTN (6/6)**
>
> # Q4 The design of the condition seems very ad-hoc
>
> Thank you for your feedback. The quantization in the condition design is indeed chosen intuitively for two reasons:
>
> 1. By quantizing the condition, we introduce an information bottleneck that prevents the model from memorizing the training data, encouraging it to capture generalizable patterns.
>
> 2. Quantization helps the model discover and encode inherent clusters of the data semantics, avoiding encoding noise or overly fine details irrelevant to the hierarchy.
>
> # Q5 Citation needed in L32
>
> Thanks for pointing out this, we have added the citation for INRs.

---

> ### Author Response · Authors · 2024-11-22
>
> Thanks for your constructive comments and valuable questions. We are happy to engage in further discussions to address any additional concerns.

---

> > ### Comment · Reviewer_LwTN · 2024-11-25
> >
> > Thank you for your rebuttal.
> > I am not sure I fully grasp the intuition for Q1. I understand the spectral bias and the guidance through hierarchy, but to think that it aligns so closely with objects (which comprise a mix of frequencies in their features) is still unintuitive to me. Cool, though!
> >
> > Otherwise, my comments have been addressed and I feel slightly more confident about the paper, thus raise my confidence score.

---

> > > ### Author Response · Authors · 2024-11-25
> > >
> > > Thanks for your time and your follow-up question!
> > >
> > > Objects like chairs and jets indeed consist of a mix of frequencies, but their low-frequency components capture the broad, coarse structures that differentiate them—e.g., a chair’s boxy, segmented structure versus a jet’s streamlined shape. These differences in global shape and spatial composition are inherently semantically meaningful and sufficient to distinguish categories at a coarse level.
> > >
> > > Our model leverages this through its spectral bias, embedding low-frequency features in early layers where global shapes dominate. The conditional chain then adds finer details layer by layer to the overall structure, preserving the broad distinctions between object categories. This enables the model to align naturally with human abstractions and generate semantically coherent outputs.
> > >
> > > We hope this further clarifies the intuition behind the close alignment with human abstraction, and hope that our clarifications solve all your concerns. We would sincerely appreciate it if you could raise your rating, enabling our observations and contributions to gain greater visibility within the community.

---

> > > > ### Comment · Reviewer_LwTN · 2024-11-26
> > > >
> > > > That intuition helps, thank you! Good luck

---

### Official Review · Reviewer_wk7N · 2024-11-01

**Soundness:** 3
**Presentation:** 4
**Contribution:** 3
**Rating:** 6
**Confidence:** 3

**Summary:**

This paper introduces a novel deep generative model for Implicit Neural Representations (INRs), designed to capture hierarchical dependencies between INR layers through a structured latent space. The proposed model, CHINR, employs a Layers of Experts (LoE) framework, where weights are globally shared, and the expert weights are obtained through data-specific learnable latent vectors mapped into gates for the LoE. After learning these latent vectors, the second stage leverages a hierarchical conditional diffusion model to approximate $p(\mathbf{h})$ in a layered manner. Extensive empirical evaluation supports the design choices, showing that CHINR outperforms most baselines while offering improved interpretability and controllable generation capabilities.

**Strengths:**

- The contributions are both relevant and well-supported by empirical results.
- The paper is clearly structured and easy to follow.
- The proposed method demonstrates robustness across evaluations.
- Limitations are thoughtfully discussed, with suggestions for future work to address them.

**Weaknesses:**

- The background section omits a few relevant recent works.

**Questions:**

### Background

- Recent generative models of INRs are not referenced in the background section. While [1-4] are included as methods that learn the latent distributions that map to or modulate INR parameters, it’s important to note that not all of these methods use meta-learning or a two-stage training approach. Specifically, [1,5,6] introduced end-to-end models where both the latent model and the latent-to-INR model are learned jointly, using GANs and VAEs. Also, there are other recent two-stage works that trained diffusion models over the latent vectors mapped to INRs, such as [7]. I suggest the authors include all the references to these additional works.

### Minor Comments

- In Section 3, the text states that "the high dimensionality of raw weights is intractable." However, in [1,5,6] the full set of INR weights is efficiently generated via hypernetworks, achieving competitive results.

- Typo on Line 465: "dependecy" should be corrected to "dependency."

### References

[1]: Dupont, Emilien, Yee Whye Teh, and Arnaud Doucet. "Generative Models as Distributions of Functions." International Conference on Artificial Intelligence and Statistics. PMLR, 2022.

[2]: Du, Y., Collins, K., Tenenbaum, J., & Sitzmann, V. (2021). Learning signal-agnostic manifolds of neural fields. Advances in Neural Information Processing Systems, 34, 8320-8331.

[3]: Dupont, Emilien, et al. "From data to functa: Your data point is a function and you can treat it like one." International Conference on Machine Learning. PMLR, 2022.

[4]: Bauer, Matthias, et al. "Spatial functa: Scaling functa to ImageNet classification and generation." arXiv preprint arXiv:2302.03130 (2023).

[5]: Koyuncu, Batuhan, et al. "Variational Mixture of HyperGenerators for Learning Distributions over Functions." International Conference on Machine Learning. PMLR, 2023.

[6]: Xu, Siyuan, et al. "Uncertainty-aware Continuous Implicit Neural Representations for Remote Sensing Object Counting." International Conference on Artificial Intelligence and Statistics. PMLR, 2024.

[7]: Park, Dogyun, et al. "DDMI: Domain-agnostic Latent Diffusion Models for Synthesizing High-Quality Implicit Neural Representations." The Twelfth International Conference on Learning Representations.

---

> ### Author Response · Authors · 2024-11-22
> **Response to Reviewer wk7N (1/2)**
>
> We appreciate Reviewer wk7N for their valuable comments.
>
> # Q1 Background missing reference to recent works
>
> Thanks for this constructive suggestion. We have included these references in our background section for a more comprehensive discussion. While [1, 2, 3, 4] are included as methods that focus on latents, we do not restrict the learning methods to meta-learning or two-stage training. However, we recognize the need for a more detailed discussion of their techniques and contributions. Specifically:
>
> - Dupont et al. [1] propose an end-to-end method using a GAN to train a hypernetwork that generates INR parameters for synthesizing new data.
>
> - Koyuncu et al. [5] extend GASP’s capabilities for efficient inference tasks (eg. in-painting) via VAE. This framework incorporates a learned prior modeled by Normalized Flow.
>
> - Du et al. [2] learn a signal-agnostic data manifold, where latents are mapped to INRs across different data modalities via different hyper-networks.
>
> - Dupont et al. [3] adopt a two-stage framework, employing meta-learning to jointly optimize a shared base INR and data-specific modulation latents.
>
> - Bauer et al. [4] extend Functa to scale to ImageNet by incorporating spatial information into the latents.
>
> - Park et al. [7] also employ a two-stage framework, where learned latents are mapped to positional embeddings (rather than neural weights) to modulate INRs and generate new data.
>
> - Xu et al. [6] focus on adapting INR parameters for object counting, where a hypernetwork uniquely tailors parameters for each input image to improve training convergence. This work emphasizes generalizing INRs to different inputs rather than generating new data.
>
> [1] Emilien Dupont, Yee Whye Teh, and Arnaud Doucet. Generative Models as Distributions of Functions, AISTATS 2022.
>
> [2] Yilun Du, Katie Collins, Josh Tenenbaum, and Vincent Sitzmann. Learning signal-agnostic manifolds of neural fields, NeurIPS 2021.
>
> [3] Emilien Dupont, Hyunjik Kim, SM Eslami, Danilo Rezende, and Dan Rosenbaum. From data to functa: Your data point is a function and you can treat it like one, ICML 2022.
>
> [4] Matthias Bauer, Emilien Dupont, Andy Brock, Dan Rosenbaum, Jonathan Richard Schwarz, and Hyunjik Kim. Spatial functa: Scaling functa to imagenet classification and generation, arXiv preprint arXiv:2302.03130, 2023.
>
> [5] Batuhan Koyuncu, Pablo Sanchez-Martin, Ignacio Peis, Pablo M Olmos, and Isabel Valera. Variational Mixture of HyperGenerators for Learning Distributions over Functions, ICML 2023.
>
> [6] Siyuan Xu, Yucheng Wang, Mingzhou Fan, Byung-Jun Yoon, and Xiaoning Qian. Uncertainty-aware Continuous Implicit Neural Representations for Remote Sensing Object Counting, AISTATS 2024.
>
> [7] Dogyun Park, Sihyeon Kim, Sojin Lee, and Hyunwoo J Kim. DDMI: Domain-agnostic Latent Diffusion Models for Synthesizing High-Quality Implicit Neural Representations, ICLR 2024.

---

> ### Author Response · Authors · 2024-11-22
> **Response to Reviewer wk7N (2/2)**
>
> # Q2 Minor comments
>
> - *"the high dimensionality of raw weights is intractable"* -- This statement in Section 3 refers to the challenges of directly learning the **distribution** of high-dimensional raw INR weights for generative modeling, not just generating weights efficiently as in [1, 2, 3]. However, we understand that this phrasing may lead to misinterpretation. We have revised it to: *"the high dimensionality of raw weights makes distribution modeling highly challenging"*
>
> - *Typo on Line 465* -- Thank you for pointing out this. We have revised it in our new submission.
>
> [1] Emilien Dupont, Yee Whye Teh, and Arnaud Doucet. Generative Models as Distributions of Functions, AISTATS 2022.
>
> [2] Batuhan Koyuncu, Pablo Sanchez-Martin, Ignacio Peis, Pablo M Olmos, and Isabel Valera. Variational Mixture of HyperGenerators for Learning Distributions over Functions, ICML 2023.
>
> [3] Siyuan Xu, Yucheng Wang, Mingzhou Fan, Byung-Jun Yoon, and Xiaoning Qian. Uncertainty-aware Continuous Implicit Neural Representations for Remote Sensing Object Counting, AISTATS 2024.

---

> ### Author Response · Authors · 2024-11-22
>
> Thanks for your valuable feedback on improving the background and presentation. We are happy to engage in further discussions. Please let us know if you have any additional concerns.

---

> ### Comment · Reviewer_wk7N · 2024-11-24
> **Rebuttal response**
>
> I thank the authors for their thorough rebuttal. All of my concerns have now been addressed, and I continue to recommend acceptance.

---

> > ### Author Response · Authors · 2024-11-25
> >
> > Thank you once again for your time and feedback. We are happy that our clarifications solve all your concerns. We would sincerely appreciate it if you could raise your score, enabling our observations and contributions to gain greater visibility within the community.

---

### Official Review · Reviewer_1yWi · 2024-11-03

**Soundness:** 3
**Presentation:** 3
**Contribution:** 2
**Rating:** 5
**Confidence:** 1

**Summary:**

In this paper, authors first posit that current generative Implicit Neural Representations (INRs) ignore hierahical structure in latent parameter distributions. To address this issues, authors propose Controllable Hierarchical Implicit Neural Representation (CHINR), consisting of two innovations: (1) Layers-of-Experts (LoE) Network and (2) Hierarchical Controllable Diffusion Model (HCDM).

**Strengths:**

1. The paper addresses a core limitation in Implicit Neural Representations (INRs): the use of a "flat" latent distribution $h$ which overlooks the inherent hierarchical structure of the data. To resolve this, the authors propose a diffusion model that leverages conditional distributions to capture this hierarchy. The motivation is clearly articulated, and the solution is well-suited to tackle the challenge by directly modeling hierarchical dependencies in the latent space.

2. The paper is well-written, presenting the proposed method with clarity and precision. Despite the complexity of the approach, the authors have structured the content effectively, making it accessible and straightforward to follow.

**Weaknesses:**

1.  The improvement over the most comparable method, mNIF, appears incremental. This paper builds upon the mNIF framework, adopting a similar two-stage process with latent modulation and a mixture of networks/layers, targeting the same challenge of capturing layer-wise hierarchical structure. However, despite the added complexity and potential inference cost, the performance gains relative to mNIF methods seem incremental. I am not sure the increased model complexity justifies the marginal improvements — since the method seems quite complicated.

2. Although the analysis in Section 4.3 effectively demonstrates how different layers control distinct semantic aspects of the generated content and highlights the importance of conditional modeling, the ablation study in Section 4.4 is less convincing. The authors primarily provide anecdotal examples without sufficient quantitative evidence. Even though the examples shown in Figure 8 indicate the importance of conditioning, expanding the experimental details and including quantitative metrics would be helpful — since it is so central to the main contribution of the paper.

**Questions:**

1. I am not an expert on this, but I noticed that mNIF reports the number of parameters and inference cost across different generation models, which seems relevant for a fair comparison. Would it be important to consider the size of the data generator as a factor here? How does CHINR's inference cost and model size compare to its competitors?

---

> ### Author Response · Authors · 2024-11-22
> **Response to Reviewer 1yWi (1/3)**
>
> We appreciate constructive comments from Reviewer 1yWi.
>
> # W1 Incremental contribution over mNIF
>
> Our proposed CHINR targets different goals from previous works, e.g. mNIF, Functa. We aim to achieve data generation controllability by modeling the conditional dependencies of model parameters, while others, such as mNIF, focus on improving generation quality. We are the first in INR's community to explore controllability by leveraging layer-wise hierarchical structure in parameter space. Moreover, our experimental analysis reveals that semantics information can be disentangled in parameter space, which hasn't ever been observed. Additionally, our work avoids trivial memorization and generates more diverse data than mNIF as presented in L357-L363 of the main paper and Figure 9 of the supplementary materials.
>
> Overall, the main contributions of our work are demonstrated across four aspects: **controllability, generalizability, versatility**, and **quality**.
> Therefore, the quality improvement is only one of our contributions.
> Specifically, our main contributions are:
>
> - **Controllability**: Our work distinguishes from previous methods in the controllability of generating fine-grain details, which has never been explored in the implicit neural representation (INR) community.
>
> - **Generalizability**: Our work can avoid trivial memorization thus generating more diverse data compared with the SOTA methods, such as mNIF. Please find the discussion and visual samples from Line 357-363 in the main paper and Figure 9 in the supplementary material, respectively.
>
> - **Versatility**: The CHINR's performance is validated across four different data domains, i.e. facial images, point clouds, NeRFs, and motions. This demonstrates the proposed hierarchical control and conditional dependency modeling is broadly applicable to data semantics with an inherent hierarchy structure.
>
> - **Quality**: Our work outperforms the SOTA methods on both the reconstruction and generation metrics, which can be seen from Table 1 in the main paper.

---

> ### Author Response · Authors · 2024-11-22
> **Response to Reviewer 1yWi (2/3)**
>
> # W2 Quantitative comparison of conditional dependency
>
> Thanks for your suggestion, we evaluate the importance of conditional dependency by progressive ablation, as shown in the table below. The experiments in the table reveal that enhancing the modeling of conditional dependency (from bottom to top) helps generate higher-quality data.
>
> The experiments start with all five layers (the top row) in the conditional chain and then progressively exclude layers from the chain. In each setting, the excluded layers are trained using unconditional diffusion models. TThe table shows that the conditional chain is essential in producing high-quality generations from hierarchical sampling, as it ensures latent compatibility across layers and alignment with the data's semantic structure.
>
> | **Layers in chain** | **Independent layers** | **FID**  |
> |----------------------|------------------------|----------|
> | 1,2,3,4,5           | None                  | 13.4     |
> | 1,2,3,4             | 5                     | 13.6     |
> | 1,2,3               | 4,5                   | 15.5     |
> | 1,2                 | 3,4,5                 | 52.8     |
> | None                | 1,2,3,4,5             | 112.7    |
>
> *Table*: Ablation on the number of layers in the conditional chain, evaluated on CelebA-HQ.

---

> ### Author Response · Authors · 2024-11-22
> **Response to Reviewer 1yWi (3/3)**
>
> # Q1 Model size & inference cost
>
> We focus on improving the controllability of data generation in INR, where model efficiency is not our primary focus. Nevertheless, we achieve better quality with a smaller number of parameters and comparable inference costs.
>
> ## Reviwer concerns
>
> > but I noticed that mNIF reports the number of parameters and inference cost across different generation models
>
> Answer: It is noted that mNIF evaluates the number of parameters and inference cost purely on the INR, not the generator (diffusion model).
>
> > Would it be important to consider the size of the data generator as a factor here?
>
> Answer: For the generator, the backbone of HCDM has the same size as mNIF. Our HCDM employs an extra $128\times12$ fully-connected layer to form the conditions and a $12\times4096$ fully-connected layer to embed binary conditions, which brings negligible model size increment.
>
> > How does CHINR's inference cost and model size compare to its competitors?
>
> Answer: We report the number of parameters, inference cost, and size of the generator in table as below.
>
> | **Model** | **# Params** | **Inference cost** fps↑ | **Memory (MB)↓** | **Generator** |
> |-----------|--------------|--------------------------|-------------------|---------------|
> | mNIF      | 33.4M        | 891.3                   | 24.4              | 2.2 GB        |
> | CHINR     | 32.5M        | 1096.4                  | 26.3              | 2.2 GB        |
>
> *Table*: Quantitative results retrained on CelebA-HQ.

---

> ### Author Response · Authors · 2024-11-22
>
> Thanks for your constructive comments and insightful questions. We are happy to engage in further discussions. Please let us know if your have any other concerns.

---

> ### Comment · Reviewer_1yWi · 2024-11-24
>
> Thank you for the detailed comments! Even if the quality improvement is incremental, the strength of the paper lies in controllability and generalizability. I think authors clarification during rebuttal makes this aspect way more clear!
> I am not an expert in this area and nor have I worked in this domain or even a related one. Therefore, I do not feel too qualified to provide any insights on the quality of this paper.  I have decreased my confidence score to reflect this, and have notified ACs to seek an opinion from different reviewers. Best of luck!

---

> > ### Author Response · Authors · 2024-11-25
> >
> > Thank you for your thoughtful feedback and for recognizing the strengths of our paper in controllability and generalizability. We’re happy our rebuttal clarified these aspects. We appreciate your detailed comments and efforts despite not being from this domain and thank you for notifying the AC to seek additional opinions.
> >
> > We hope our answers help provide clarity and highlight the contributions of our work to other reviewers and readers. Best regards!

---

### Official Review · Reviewer_gfwF · 2024-11-03

**Soundness:** 3
**Presentation:** 3
**Contribution:** 2
**Rating:** 5
**Confidence:** 4

**Summary:**

This paper proposes a novel method for Implicit Neural Representations (INRs) called Controllable Hierarchical Implicit Neural Representation (CHINR). The method introduces a hierarchical approach to sampling and constructing latents that captures the layer-wise dependencies in the weights of a network and allows for controllable generation. The method is benchmarked on data generation against previous INR models, and qualitative results show the efficacy of the hierarchical approach.

**Strengths:**

-The hierarchical approach is interesting and worth exploring in more detail.

-The diffusion approach is novel and seems like a good approach to modeling the causal structure between the latents across different layers.

**Weaknesses:**

Overall, the paper seems incomplete to me. Table 1 is missing a lot of values, and there is a lack of benchmarking. The idea is not wholly uninteresting—in fact, I think with more benchmarking on the novel aspects of their method, the authors can put together a compelling manuscript. Ultimately, however, this manuscript is not ready for ICLR. I suggest the authors use this feedback to prepare for a submission to the next available proceedings.

-The glaring weakness in this paper is the lack of benchmarking against alternative methods. Even the only benchmarking table, Table 1, is incomplete. The only method that was fully benchmarked is mNIF, which seems to be the predecessor upon which this paper is based. Furthermore, the generation results show only marginal improvement, and it is hard to determine whether these are significant without repeats and standard errors. E.g. line 354: “Our model outperforms existing methods on most datasets.”—this is not sufficiently justified by the half-completed table.

-Related to the previous point, the hierarchical component of the model is the heart of the method, yet the only analyses I see are the qualitative figures and the correlation matrix between layers. The authors should think of a benchmark that can exploit the hierarchical structure of their model to demonstrate its utility.

-Without some kind of benchmark of the utility of the hierarchical construction of latents, the method presents itself as only a slight generalization of mNIF with much more baggage, owing to the diffusion component needed to generate the hierarchical latents. Additionally, the generation benchmark shows CHINR is at best comparable to mNIF.

-The paper is a bit hard to read—for example, I recommend clearly outlining the training procedure in stage-1, as I don’t think the mNIF method is a well-known standard method. This is less of a concern than the previous points.

**Questions:**

See the weaknesses above.

---

> ### Author Response · Authors · 2024-11-22
> **Response to Reviwer gfwF (1/4)**
>
> We thank Reveiwer gfwF for the constructive comments.
> # W1.1 Blank cells in Table 1
>
> In Table 1 of the main paper, for fair comparison, we adopt only the reported results of the existing methods. Moreover, the missing results (blank cells) come from:
>
> 1. Some existing methods cannot deal with all three datasets. For example, GASP cannot work on the NeRF data, such as SRN-Cars, because GASP requires the ground-truth radiance fields for training, which are unavailable.
>
> 2. Some existing methods are not evaluated on all the three datasets. For example, GEM is not evaluated on the NeRF data.
>
> 3. Some existing methods report only the qualitative results without the quantitative ones. For example, Functa only provides visual samples on Shapenet generation without quantitative results evaluated by the metrics.
>
> Nevertheless, we understand your concern and address it by retraining these methods under identical settings. To measure the standard deviation, each model is retrained four times with different random seeds. The results for CelebA-HQ are reported in:
>
>
> | **Model** | PSNR↑ | SSIM↑         | FID↓ |
> |-----------|---------------------------|---------------|----------------------|
> | Functa [1]    | 26.2 ± 0.3               | 0.795 ± 0.015 | 41.0 ± 0.2          |
> | GEM [2]      | 26.5 ± 0.4               | 0.814 ± 0.018 | 30.8 ± 0.3          |
> | GASP [3]     | 31.6 ± 0.8               | 0.902 ± 0.021 | 13.6 ± 0.3          |
> | mNIF [4]     | 34.5 ± 0.2               | 0.957 ± 0.005 | **13.2 ± 0.1**          |
> | CHINR     |  **34.9 ± 0.3**       |  **0.964 ± 0.006**| 13.4 ± 0.1        |
>
> The results for ShapeNet are reported in:
>
> | **Model** |  PSNR↑ | Accuracy↑      |  Coverage↑ | MMD↓             |
> |-----------|---------------------------|----------------|---------------------------|------------------|
> | Functa    | 22.1 ± 0.3               | 0.983 ± 0.005 | 0.437 ± 0.005            | 0.0013 ± 0.0002 |
> | GEM       | 21.4 ± 0.4               | 0.977 ± 0.007 | 0.408 ± 0.003            | 0.0016 ± 0.0003 |
> | GASP      | 16.7 ± 0.8               | 0.928 ± 0.011 | 0.343 ± 0.009            | 0.0023 ± 0.0006 |
> | mNIF      | 21.4 ± 0.3               | 0.975 ± 0.008 | 0.435 ± 0.003            | 0.0014 ± 0.0003 |
> | CHINR     | **22.3 ± 0.2**               | **0.988 ± 0.005** | **0.441 ± 0.002**            | **0.0011 ± 0.0001** |
>
> The results for SRN-Cars are reported in:
>
> | **Model** |  PSNR↑ | SSIM↑         |  FID↓ |
> |-----------|---------------------------|---------------|----------------------|
> | Functa    | 24.3 ± 0.2               | 0.738 ± 0.009 | 80.1 ± 0.2          |
> | mNIF      | 26.0 ± 0.3               | 0.763 ± 0.013 | 79.3 ± 0.3          |
> | CHINR     | **26.3 ± 0.2**               | **0.780 ± 0.011** | **77.8 ± 0.2**          |
>
> # W1.2 Marginal quality improvement
>
> The main contributions of our work are demonstrated across four aspects: **controllability, generalizability, versatility**, and **quality**.
> Therefore, the quality improvement is only one of our contributions.
> Specifically, our main contributions are:
>
> - **Controllability**: Our work distinguishes from previous methods in the controllability of generating fine-grain details, which has never been explored in the implicit neural representation (INR) community.
>
> - **Generalizability**: Our work can avoid trivial memorization thus generating more diverse data compared with the SOTA methods, such as mNIF. Please find the discussion and visual samples from Line 357-363 in the main paper and Figure 9 in the supplementary material, respectively.
>
> - **Versatility**: The CHINR's performance is validated across four different data domains, i.e. facial images, point clouds, NeRFs, and motions. This demonstrates the proposed hierarchical control and conditional dependency modeling is broadly applicable to data semantics with an inherent hierarchy structure.
>
> - **Quality**: Our work outperforms the SOTA methods on both the reconstruction and generation metrics, which can be seen from Table 1 in the main paper.
>
> Moreover, the analysis presented in Section 4.3 of the main paper further reveals how each layer of CHINR governs disentangled data semantics, which mathematically applies to other sinusoid-activated INR models. This property sheds new lights on the inductive bias of INR architecture.
>
> [1] Emilien Dupont, Hyunjik Kim, SM Eslami, Danilo Rezende, and Dan Rosenbaum. From data to functa: Your data point is a function and you can treat it like one, ICML 2022.
>
> [2] Yilun Du, Katie Collins, Josh Tenenbaum, and Vincent Sitzmann. Learning signal-agnostic manifolds of neural fields, NeurIPS 2021.
>
> [3] Emilien Dupont, Yee Whye Teh, and Arnaud Doucet. Generative models as distributions of functions, AISTATS 2022.
>
> [4] Tackgeun You, Mijeong Kim, Jungtaek Kim, and Bohyung Han. Generative neural fields by mixtures of neural implicit functions, NeurIPS 2024.

---

> > ### Author Response · Authors · 2024-11-22
> > **Response to Reviwer gfwF (2/4)**
> >
> > # W2 Benchmark the hierarchy
> >
> > As there is no available metric to benchmark the utility of the hierarchical structure in the INR literature, we chose not to include a benchmark in the main paper.
> > However, we understand your concerns and appreciate your suggestions for quantitative benchmarking. Therefore, we benchmark that the hierarchical structure improves controllability in the table, showing that hierarchical structure enables controlling semantics at different levels.
> >
> > | **Attributes**   | **Random** | **L1**     | **L2**     | **L3**     | **L4**     |
> > |-------------------|------------|------------|------------|------------|------------|
> > | Oval face        | 30.3%      | **73.2%**  | 85.7%      | 90.2%      | 94.1%      |
> > | Blonde hair      | 39.6%      | 40.5%      | **85.8%**  | 91.4%      | 96.8%      |
> > | Smiling          | 30.8%      | 31.2%      | 33.6%      | **83.0%**  | 92.5%      |
> > | Red lips         | 10.5%      | 11.3%      | 13.7%      | 12.5%      | **95.3%**  |
> >
> > *Table*: Column *random*: The positive ratio of attributes in random samples (no control). Columns *L1-4*: The success rate of attribute control when fixing latents up to a specific layer and sampling from subsequent layers.
> >
> > We begin by randomly generating $1,000$ facial images and calculating the ratio of samples where each attribute (as shown in table) is positive. For each attribute, we then take the positive samples and progressively fix the latents of the initial layers while sampling the remaining layers. At each step, we compute the positive ratio of the newly generated samples to verify whether the attribute is preserved. This experiment assesses the model's ability to maintain control over fixed attributes while varying others. The table demonstrates the effectiveness of coarse-to-fine, layer-specific control enabled by the latent hierarchy. The bold numbers denote a significant increase in success rate compared to the previous layer, showing that the corresponding attributes are effectively controlled at that layer.

---

> > > ### Author Response · Authors · 2024-11-22
> > > **Response to Reviwer gfwF (3/4)**
> > >
> > > # W3 Slight generalization over mNIF
> > >
> > > Our method aims to achieve control over data generation by modeling the conditional dependencies of model parameters, whereas others, such as mNIF, primarily emphasize improving generation quality. Overall, our contributions are highlighted in four aspects: **controllability, generalizability, versatility**, and **quality**. While the quality improvement over the mNIF, Functa, GASP, and GEM validates the effectiveness of our method, the controllability, generalizability, and versatility stand out as the crowning achievements.

---

> > > > ### Author Response · Authors · 2024-11-22
> > > > **Response to Reviwer gfwF (4/4)**
> > > >
> > > > # W4 The presentation is hard to follow
> > > >
> > > > We believe our presentation including the challenges, motivations, insightful intuitions, solutions, and experiments is organized efficiently. This is also the opinion of other reviewers. We appreciate the suggestions for adding training details for stage 1 and will add them to the supplementary materials.

---

> ### Author Response · Authors · 2024-11-22
>
> Thanks for your constructive comments and insightful questions. We are happy to engage in further discussions. Please let us know if your have any other concerns.

---

> > ### Author Response · Authors · 2024-11-25
> >
> > Thank you again for your time and efforts.
> >
> > The deadline for author-reviewer discussions is approaching. If you still need any clarification or have any other questions, please do not hesitate to let us know. If you are satisfied with our responses, we would greatly appreciate it if you could reevaluate the manuscript and update your rating accordingly.

---

> ### Comment · Reviewer_gfwF · 2024-12-03
>
> Thank you for the additional experiments and clarifications.
>
> The additional benchmarks add some robustness to the method's performance relative to other models, and the table showing the hierarchical analysis adds strength to the claims of controllability.
>
> Overall, the performance of the method is still only comparable to mNIF. Figure 9 does indeed show better generalization, but I assume it is curated without further explication in the caption or text below the caption.
>
> Moving past the benchmarks, the approach to controllable generation and modeling hierarchical features does seem to be somewhat novel. While modeling hierarchical features itself is not new to the field [1] (thanks to reviewer wk7N for pointing out this paper) or [2] for example, the usage of Layer-of-Experts seems to be new. However, the diffusion-generated latents is not novel [3]. Therefore, we are left with looking to the Layer-of-Experts in combination with its effect on hierarchical generation.
>
> In my opinion, the novelty and contribution does not outweigh the lack of clear improvement and/or benefit from CHINR's hierarchical approach. I will raise my score to a 5 for the authors' additional work in addressing my concerns.
>
> ### **References**
> [1] Park, Dogyun, et al. "DDMI: Domain-agnostic Latent Diffusion Models for Synthesizing High-Quality Implicit Neural Representations." The Twelfth International Conference on Learning Representations.
>
> [2] Hao, Zekun, et al. "Implicit Neural Representations with Levels-of-Experts" NeurIPS, 2022
>
> [3] Erkoç, Ziya, et al. "Hyperdiffusion: Generating implicit neural fields with weight-space diffusion" Proceedings of the IEEE/CVF International Conference on Computer Vision (ICCV), 2023

---

> ### Author Response · Authors · 2024-12-03
>
> Thank you for your valuable feedback! We are happy that our responses addressed your concerns regarding CHINR’s robustness and controllability. Below, we clarify your remaining points.
>
> ## Point 1
> > Overall, the performance of the method is still only comparable to mNIF.
>
> As replied in our general response and responses to reviewers (“Response to Reviewer gfwF (1/4): W1.2,” “Response to Reviewer 1yWi (1/3),” and “Response to Reviewer LwTN (1/6): W1-Strengths”), our work outperforms other methods across four key aspects: **controllability, generalizability, versatility**, and **quality**. **In particular, our novel controllable data generation has been recognized by Reviewer 1yWi, wk7N, LWtN, and ww4M.**
>
> In particular, CHINR is designed with distinct goals compared to prior works such as mNIF. While these methods primarily focus on improving quality metrics, CHINR aims to achieve controllability by modeling conditional dependencies in the parameter space. To our knowledge, we are the first to explore controllability by leveraging a layer-wise hierarchical structure in the INR community. This hierarchical control mechanism is particularly valuable in scenarios requiring fine-grained modifications, an aspect lacking in mNIF and similar methods.
>
> ## Point 2
> > Figure 9 does indeed show better generalization, but I assume it is curated without further explication in the caption or text below the caption.
>
> Thank you for recognizing CHINR’s generalizability. We want to clarify that Figure 9 is not curated but rather randomly generated and retrieved from the training set. The process for retrieval is described in detail in the text below the caption.
>
> ## Point 3
> > While modeling hierarchical features itself is not new to the field [1] (thanks to reviewer wk7N for pointing out this paper) or [2] for example, the usage of Layer-of-Experts seems to be new.
>
> We appreciate your acknowledgment of the novelty in our Layer-of-Experts. However, we would like to emphasize the key distinctions from [1] and [2]:
> 1. Our work does not model the **hierarchical features** but instead focuses on the **hierarchy of INR parameters**. This distinction highlights a new direction in exploring semantic hierarchy in parameter space.
> 2. Our work is new to the INR community in modeling the semantics hierarchy by capturing conditional dependencies across layer-wise parameters.
> 3. Different Goals: The works cited ([1] and [2]) aim primarily to improve quality, while CHINR focuses on controllable data generation.
> 4. Architectural Differences
>     * [1]: This work uses layer-wise positional embeddings without modeling their conditional dependencies. In contrast, we model conditional dependencies through layer-wise modulation latents.
>     * [2]: While its levels-of-experts overfit each data instance with spatially varying neural weights, CHINR generalizes to different data instances with combinations of experts in a spatially invariant way.
>
> ## Point 4
> > However, the diffusion-generated latents is not novel [3].
>
> We did not claim as a novelty to use diffusion-generated latents. We have already presented in background section (L133-L134) that the referred work [3] generates latents by a diffusion model.
>
> Our novelty about diffusion is to design an HCDM that enables control over different semantic granularities. It is also a recognized strength in your previous comments ("Official Review of Submission5442 by Reviewer gfwF") and other reviewers' comments ("Official Review of Submission5442 by Reviewer 1yWi" and "Official Review of Submission5442 by Reviewer ww4M").
>
> ## Point 5
> > In my opinion, the novelty and contribution does not outweigh the lack of clear improvement and/or benefit from CHINR's hierarchical approach.
>
> We respectfully disagree and reiterate that our primary objective is to improve controllability in data generation, as reflected in our title.
>
> Thanks again for your valuable feedback. We hope these clarifications address your remaining concerns. We are happy to engage in further discussions. If you find that all your concerns have been resolved, we would sincerely appreciate your consideration in raising your score.
>
> **References**
>
> [1] Park, Dogyun, et al. "DDMI: Domain-agnostic Latent Diffusion Models for Synthesizing High-Quality Implicit Neural Representations." The Twelfth International Conference on Learning Representations.
>
> [2] Hao, Zekun, et al. "Implicit Neural Representations with Levels-of-Experts" NeurIPS, 2022
>
> [3] Erkoç, Ziya, et al. "Hyperdiffusion: Generating implicit neural fields with weight-space diffusion" Proceedings of the IEEE/CVF International Conference on Computer Vision (ICCV), 2023

---

> > ### Author Response · Authors · 2024-12-03
> >
> > Thank you once again for your time and thoughtful feedback!
> >
> > As the deadline for author-reviewer discussions approaches, please don’t hesitate to reach out if you need any further clarification or have additional questions.

---

> ### Author Response · Authors · 2024-12-03
>
> Dear Reviewer gfwF,
>
> As the deadline for author-reviewer discussions approaches, please don’t hesitate to reach out if you need any further clarification or have additional questions.
>
> Best,
>
> Authors

---

### Author Response · Authors · 2024-11-22
**General response**

We sincerely  thank all reviewers for their constructive comments and insightful questions. We are happy to see all reviewers recognize our novelty in improving controllability of generative INRs by conditional dependency modeling. We have provided detailed responses to each comment to clarify ambiguities and highlight our contributions. The revised parts are marked in blue in the main paper.

Before answering each comment, we would like to clarify a few keypoints:

## Contributions

The main contributions of our work are demonstrated across four aspects: **controllability, generalizability, versatility**, and **quality**, which are beyond a slight generalization of existing works.
Specifically, our main contributions are:

- **Controllability**: Our work distinguishes from previous methods in the controllability of generating fine-grain details, which has never been explored in the implicit neural representation (INR) community.

- **Generalizability**: Our work can avoid trivial memorization thus generating more diverse data compared with the SOTA methods, such as mNIF. Please find the discussion and visual samples from Line 357-363 in the main paper and Figure 9 in the supplementary material, respectively.

- **Versatility**: The CHINR's performance is validated across four different data domains, i.e. facial images, point clouds, NeRFs, and motions. This demonstrates the proposed hierarchical control and conditional dependency modeling is broadly applicable to data semantics with an inherent hierarchy structure.

- **Quality**: Our work outperforms the SOTA methods on both the reconstruction and generation metrics, which can be seen from Table 1 in the main paper.

## Conditional depency modeling

We thank Reviwers gfwF and 1yWi for insghtful questions about evaluating the hierarchy structure of the model, which we demonstrate through qualitative results in main paper. We recognize their suggestions on quantitative results as a valuable supplement for clarification of our contributions, and include the detailed discussions in "Response to Reviwer gfwF (2/4)" and "Response to Reviewer 1yWi (2/3)".

We thank Reviwer LwTN for the insightful questions about discrepency in conditional chain. We provide detailed discussions in "Response to Reviewer LwTN (2/6)" on how we address this concern by conditonal sampling and our designed training and inference strategies.

We hope our responses address your concerns. We are happy to engage in further discussions. Please let us know if you have any other questions.

---

### Meta-Review · Area_Chair_NWGi · 2024-12-24

**Metareview:**

The paper introduces a framework for capturing hierarchical data semantics in implicit neural representation (INR) models, aiming to generate images with improved control over expressivity and diversity. The framework consists of two main components. First, multilayer INRs are trained on a dataset, with each layer parameterized by a mixture of experts and a latent vector learned through auto-decoding. This approach results in a flexible model. Second, a distribution over the latent representations is learned, factorized in a hierarchical structure where the distribution at layer k is conditioned on the latent distributions of layers <k. This distribution is modeled as a hierarchical diffusion process. The motivation is that each layer captures specific semantics (e.g., frequency features) of an image. Once trained, the model generates images by sampling from the latent distribution space. The model is evaluated on four image datasets and compared quantitatively to baselines in terms of reconstruction ability and qualitatively in its capacity to generate realistic and diverse data by leveraging the learned hierarchical distribution.

The main novelty of the paper lies in its hierarchical component and the conditional latent distribution. While quantitative evaluations demonstrate state-of-the-art (SOTA) performance in reconstruction and generation, the improvements are incremental compared to simpler methods that do not model hierarchical INR dependencies, as highlighted by the reviewers. Thus, the primary contribution of the paper lies in its potential for better control over pattern generation. Reviewers acknowledged this novelty and recognized the significance of generating diverse, high-quality images with controlled attributes. They also appreciated the authors' efforts during the rebuttal process. However, reviewers noted that the qualitative analysis did not fully demonstrate the model's ability to capture different levels of abstraction effectively. While it is understandable that this aspect is more challenging to evaluate than plain reconstruction, the authors are encouraged to strengthen this part of their contribution in future work.

**Additional Comments On Reviewer Discussion:**

Following the reviewers' requests for quantitative evaluations of reconstruction and generation, the authors added new experiments and provided clarifications. However, after the rebuttal, opinions remained mixed, with some reviewers unconvinced by the demonstration and analysis of controllable generation.

---

### Decision · Program_Chairs · 2025-01-22

Reject